# Impaired mitochondrial calcium efflux contributes to disease progression in models of Alzheimer's disease

Pooja Jadiya [1], Devin W. Kolmetzky[1], Dhanendra Tomar [1], Antonio Di Meco[1,2], Alyssa A. Lombardi[1], Jonathan P. Lambert [1], Timothy S. Luongo[1], Marthe H. Ludtmann[3], Domenico Praticò[1,2] & John W. Elrod [1]

Impairments in neuronal intracellular calcium ($_iCa^{2+}$) handling may contribute to Alzheimer's disease (AD) development. Metabolic dysfunction and progressive neuronal loss are associated with AD progression, and mitochondrial calcium ($_mCa^{2+}$) signaling is a key regulator of both of these processes. Here, we report remodeling of the $_mCa^{2+}$ exchange machinery in the prefrontal cortex of individuals with AD. In the 3xTg-AD mouse model impaired $_mCa^{2+}$ efflux capacity precedes neuropathology. Neuronal deletion of the mitochondrial $Na^+/Ca^{2+}$ exchanger (NCLX, *Slc8b1* gene) accelerated memory decline and increased amyloidosis and tau pathology. Further, genetic rescue of neuronal NCLX in 3xTg-AD mice is sufficient to impede AD-associated pathology and memory loss. We show that $_mCa^{2+}$ overload contributes to AD progression by promoting superoxide generation, metabolic dysfunction and neuronal cell death. These results provide a link between the calcium dysregulation and metabolic dysfunction hypotheses of AD and suggest $_mCa^{2+}$ exchange as potential therapeutic target in AD.

---

[1] Center for Translational Medicine, Department of Pharmacology, Lewis Katz School of Medicine at Temple University, Philadelphia, PA 19140, USA. [2] Alzheimer's Center at Temple, Lewis Katz School of Medicine at Temple University, Philadelphia, PA 19140, USA. [3] Royal Veterinary College, 4 Royal College Street, Kings Cross, London, UK. Correspondence and requests for materials should be addressed to J.W.E. (email: elrod@temple.edu)

  1

Alzheimer's disease (AD) is an age-associated multifactorial disease characterized by two major pathological hallmarks, senile plaques composed of extracellular β-amyloid (Aβ) and neurofibrillary tangles (NFTs) composed of hyperphosphorylated tau. These protein aggregates are thought to trigger neurodegeneration, resulting in loss of memory and cognitive skills in both sporadic and familial forms of AD[1]. Over the past two decades, there has been progress in identifying various cellular mechanisms contributing to AD pathogenesis, but at present the underlying cause is unknown and thus far therapeutic strategies targeting Aβ and tau pathways have failed[2,3].

It is widely accepted that intracellular calcium ($_iCa^{2+}$) signaling has an essential role in synaptic transmission and neuronal intra- and paracellular communication. The majority of neuronal ATP is consumed to modulate ion flux at the plasma membrane and ER to control $_iCa^{2+}$ levels[4].

Given the essential nature of $Ca^{2+}$ regulation to cellular homeostasis, it is not surprising that alterations in $Ca^{2+}$ handling have been reported to be a central feature of neurodegeneration and age-related diseases[5]. Numerous reports of $Ca^{2+}$ dysregulation coalesced into the formation of the "calcium hypothesis of aging and AD"[6,7]. The calcium hypothesis theorizes that alterations in $Ca^{2+}$ handling, leading to elevations in $_iCa^{2+}$, are a central mechanism linking amyloid metabolism to neuronal cell death and cognitive decline. Indeed, numerous molecular mechanisms have been shown to contribute to amyloid-mediated impairments in $Ca^{2+}$ regulation at multiple cellular levels including: altered SERCA activity, increased ryanodine receptor leak[8], and increased inositol 1,4,5-trisphosphate receptor activity at the ER[9], the dysregulation of voltage-operated channels such as calcium homeostasis modulator 1[10], altered receptor-dependent $Ca^{2+}$ signaling (nicotinic acetylcholine receptors, N-methyl-D-aspartate receptors[7] and amino-3-hydroxy-5-methyl-4-isoxazolepropionic acid receptors[11] and the dysregulation of store-operated calcium entry at the plasma membrane[12]. Inversely, there is growing evidence that $Ca^{2+}$ dysregulation is upstream in the disease process and perhaps precede amyloidogenic disease. This has prompted some investigators to propose that impairments in $Ca^{2+}$ regulation may actually drive AD development[13,14]. Regardless, there is clear evidence of $Ca^{2+}$ dysregulation with numerous studies suggesting neurons are subjected to elevated $_iCa^{2+}$ levels in AD.

This increase in $_iCa^{2+}$ coupled with alterations in ER/mitochondria tethering,[15,16] is theorized to cause mitochondrial calcium ($_mCa^{2+}$) overload, which is linked to: excessive reactive oxygen species (ROS) generation, metabolic derangement, and mitochondrial dysfunction, all prominent features of AD pathogenesis. In addition, it is widely recognized that $_mCa^{2+}$ can directly influence cell death signaling by activating the mitochondrial permeability transition pore and $Ca^{2+}$ dependent proteases (calpains), and indirectly through its effects on superoxide generation and ATP availability[17–20]. Although several studies have proposed that mitochondrial dysfunction is a primary trigger for AD development[17,21] to date no causal studies have directly implicated alterations in $_mCa^{2+}$ exchange with AD or examined the direct contribution of $_mCa^{2+}$ signaling with disease progression.

The mitochondrial $Na^+/Ca^{2+}$ exchanger (NCLX) is the primary mechanism for $_mCa^{2+}$ efflux in excitable cells[22,23], and thereby is an excellent target to modulate $_mCa^{2+}$ load in neurons. Here, we provide evidence that $_mCa^{2+}$ efflux is impaired in an age-dependent fashion in multiple experimental models of AD and that NCLX expression is lost in the frontal cortex of non-familial/sporadic AD patients. Our data suggest that the dysregulation of mitochondrial calcium efflux precedes neuropathology and memory decline in AD. We propose that these results

provide a missing link between the "calcium dysregulation" and "mitochondrial dysfunction" hypotheses and advocate targeting mitochondrial calcium exchange as a powerful therapeutic to inhibit or reverse AD progression.

## Results

**Loss of NCLX expression correlates with AD progression.** Frontal cortex samples were collected postmortem from non-familial, sporadic AD patients, and age-matched controls with no history of dementia and probed for alterations in the expression of mitochondrial calcium handling genes. Interestingly, we observed a substantial reduction in the expression of NCLX, the primary mediator of $_mCa^{2+}$ efflux in excitable cells (Fig. 1a). In addition, we observed a reduction in the mitochondrial calcium uniporter channel (mtCU)-associated proteins: MICU1 and MCUB. The voltage-dependent anion channel (VDAC) and Complex V-Sα (CV-Sα) served as mitochondrial loading controls. Next, we examined $_mCa^{2+}$ transporter expression in a well characterized, robust murine model of AD to determine whether these alterations preceeded or coincided with neurohistopathology and behavior deficits. We acquired mutant mice harboring three gene mutations associated with familial AD (3xTg-AD): Presenilin 1 (Psen1, M146V homozygous knock-in), amyloid beta precursor protein (APPswe, KM670/671NL transgene) and microtubule associated protein tau (MAPT, P301L transgene). These mice develop age-progressive pathology similar to that observed in AD patients including: impaired synaptic transmission, Aβ deposition, plaque/tangle histopathology, and learning/memory deficits beginning ~6 months of age[24]. mRNA and protein were isolated from brain tissue derived from the frontal cortex and hippocampus of 2, 4, 8, and 12-month-old 3xTg-AD mutant mice and outbred age-matched, non-transgenic controls. 3xTg-AD mice displayed an age-dependent reduction in NCLX expression with a significant decrease noted as early as 4 months and near complete loss of mRNA and protein by 12 months of age (Fig. 1b, c; Supplementary Fig. 1A–D). Importantly, we found no significant alteration in the expression of any transporters associated with $_mCa^{2+}$ exchange in samples isolated from the brains of 2-month-old 3xTg-AD mice, an age prior to any detectable neuropathology or memory deficits[25] (Fig. 1b; Supplementary Fig. 1a, e). These results suggest the changes in gene expression are age-dependent and not merely the result of developmental expression changes associated with this mutant model. The loss of the key $_mCa^{2+}$ efflux mediator, NCLX, and decrease in the expression of negative regulators of the mtCU, MICU1 (gatekeeper of mtCU to limit uptake at low $_iCa^{2+}$ and augmenter of uptake at high $_iCa^{2+}$ levels[26–29]) and MCUB[30], is likely to promote increased $[_mCa^{2+}]$ levels; especially in the high $_iCa^{2+}$ environment that is reported to occur in neurons during AD progression[6]. These alterations are in stark contrast to the compensatory alterations we have previously reported in cardiac biopsies isolated from failing hearts at the time of transplant[22], and suggest that in AD, changes in the expression profile of $_mCa^{2+}$ transporters may contribute to matrix $Ca^{2+}$ overload and disease development.

**Rescue of NCLX expression corrects impairments in $_mCa^{2+}$ exchange.** Next, we moved to a system more amendable to real-time mechanistic studies and employed a neuroblastoma cell line (N2a) stably expressing the familial APPswe mutation (K670N, M671L, APPswe[31]) coupled with a neuronal maturation protocol[32]. Surprisingly, maturated APPswe cells also displayed a significant reduction in the expression of NCLX, mirroring the results obtained from human AD brains (Fig. 1d, e). No change in VDAC or OxPhos component expression was observed,

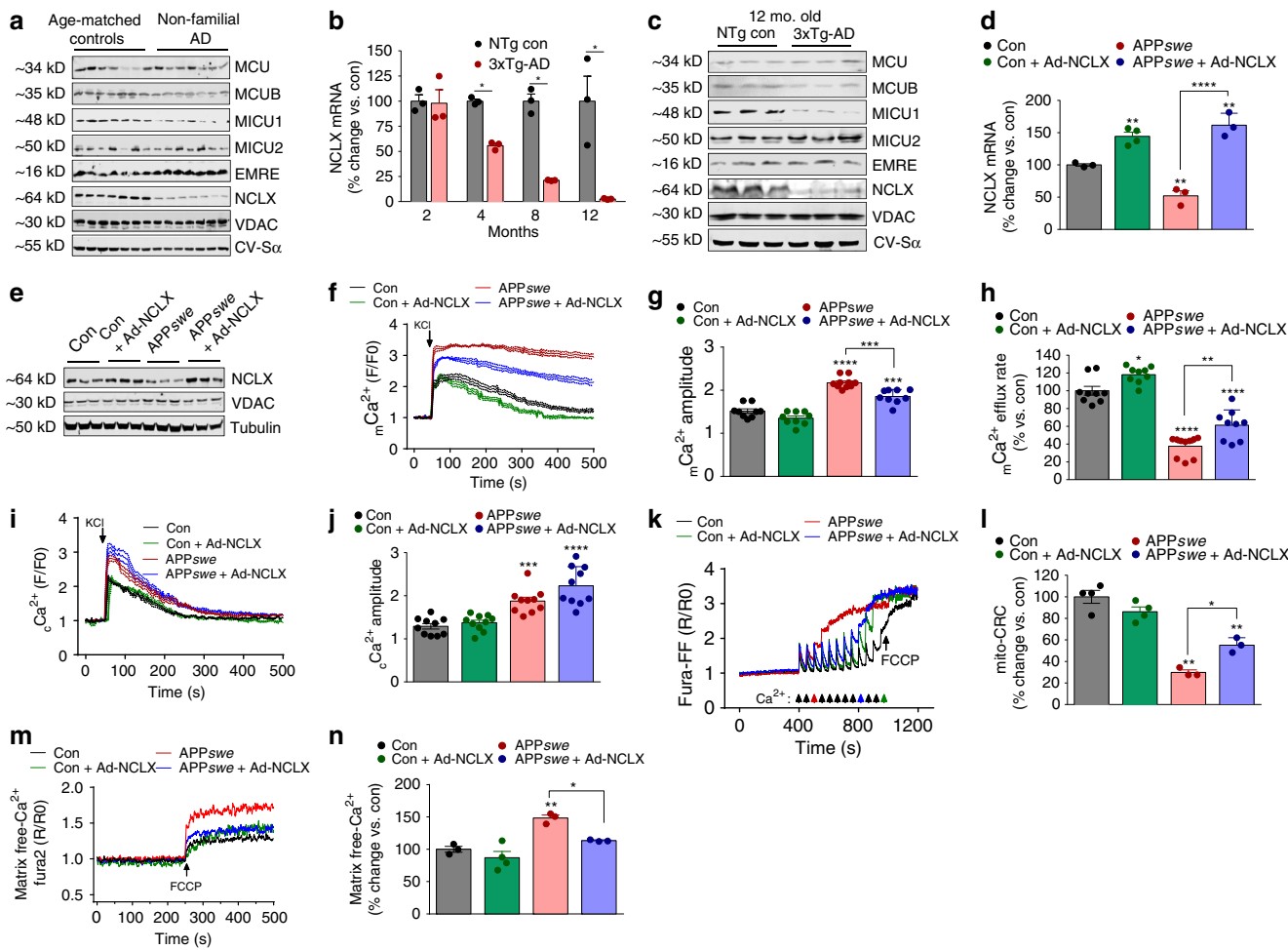

**Fig. 1** NCLX expression and mtCU components are significantly altered in AD. **a** Western blots for proteins associated with $_mCa^{2+}$ exchange in postmortem brains of patients diagnosed with non-familial, sporadic AD, and age-matched controls, $n = 7$ for both groups. MCU, Mitochondrial Calcium Uniporter; MCUB, Mitochondrial Calcium Uniporter β-subunit; MICU1, Mitochondrial Calcium Uptake 1; MICU2, Mitochondrial Calcium Uptake 2; EMRE, Essential MCU Regulator; NCLX, Mitochondrial $Na^+/Ca^{2+}$ Exchanger. Voltage-dependent anion channel (VDAC) and oxidative phosphorylation component, Complex V α-subunit (CV-Sα) were used as mitochondrial loading controls. **b** NCLX mRNA expression in tissue isolated from the frontal cortex of 3xTg-AD mutant mice and age-matched outbred non-transgenic controls (NTg). **c** Western blots for expression of $_mCa^{2+}$ exchanger in 12-month-old 3xTg-AD mutant mice and age-matched NTg controls, $n = 3$ for both groups. **d** NCLX mRNA expression in N2a control + Ad-NCLX, APPswe and APPswe + Ad-NCLX corrected to N2a control. **e** Western blots for NCLX expression in N2a control, Control + Ad-NCLX, APPswe, and APPswe + Ad-NCLX; representative of three independent experiments. **f** Mitochondrial $Ca^{2+}$ transients, mean shown as solid line, thin lines display ± SEM; $n = 9$ for Con, $n = 9$ for Con + Ad-NCLX, $n = 11$ APPswe and $n = 9$ for APPswe + Ad-NCLX. **g** Quantification of $_mCa^{2+}$ transient peak amplitude. **h** Percent $_mCa^{2+}$ efflux rate vs. control. **i** Cytosolic $Ca^{2+}$ transients, mean shown as solid line, thin lines display ± SEM. **j** Quantification of cytosolic $Ca^{2+}$ peak amplitude. **k** Representative recordings of mitochondrial $Ca^{2+}$ retention capacity. **l** Percent change in $_mCa^{2+}$ retention capacity vs. N2a control cells. **m** Representative traces for basal mitochondrial $Ca^{2+}$ content. **n** Quantification of $_mCa^{2+}$ content. ($n$ = individual dots shown for each group in all graphs. All data presented as mean ± SEM; ****$p < 0.001$, **$p < 0.01$, *$p < 0.05$; one-way ANOVA with Sidak's multiple comparisons test.) Source data are available as a Source Data file

suggesting no change in overall mitochondrial content (Fig. 1e; Supplementary Fig. 1F). To evaluate if restoring NCLX expression was sufficient to rescue impairments in $_mCa^{2+}$ handling we infected APPswe cells with adenovirus encoding NCLX (Ad-NCLX). The decrease in NCLX expression in APPswe cells was completely rescued 48 h post transduction with Ad-NCLX (Fig. 1d, e). To evaluate if the alterations in $_mCa^{2+}$-associated gene expression impacted intracellular $Ca^{2+}$ handling we examined $_cCa^{2+}$ and $_mCa^{2+}$ transients. Neurons were transduced with adeno-associated virus encoding the mitochondrial-targeted $Ca^{2+}$ reporter, R-GECO1 (AAV6-$_{mito}$R-GECO) and loaded with the $_cCa^{2+}$ reporter, Fluo4-AM and imaged continuously during stimulation with KCl to induce plasma membrane depolarization and activation of voltage-gated $Ca^{2+}$ channels (Fig. 1f, i). APPswe

cells displayed a significant increase, ~40%, in $_mCa^{2+}$-transient peak amplitude as compared to N2a control cells, and this was significantly reduced by rescuing NCLX expression (Fig. 1g). Quantification of the $_mCa^{2+}$ efflux rate revealed ~60% decrease in APPswe cells, as compared with N2a con cells, and infection with Ad-NCLX significantly restored the efflux rate (Fig. 1h). Measurements of $_cCa^{2+}$ flux confirmed elevated $_cCa^{2+}$ levels in APPswe cells, which was not impacted by rescuing NCLX expression (Fig. 1j; Supplementary Fig. 1H, I). To evaluate if impaired $_mCa^{2+}$ efflux may contribute to $_mCa^{2+}$ overload, we measured the $_mCa^{2+}$ retention capacity using the ratiometric reporters FuraFF, to monitor $Ca^{2+}$, and JC1, to monitor mitochondrial membrane potential, Δψ (Supplementary Fig. 1J–M). APPswe cells rapidly underwent permeability transition (3rd

10 µM delivery of bath $Ca^{2+}$, Fig. k, l). This was in striking contrast to control cells, which sustained three-times the concentration of bath $Ca^{2+}$ before collapse of $\Delta\psi$ and loss of matrix $Ca^{2+}$. Rescue of NCLX expression in APPswe neurons greatly increased the mitochondrial calcium retention capacity (CRC; ~9th 10 µM bath $Ca^{2+}$ injection, Fig. 1k, l). To evaluate if restoring NCLX-mediated efflux was sufficient to reduce matrix $Ca^{2+}$ load, cells were loaded with Fura2 and treated with digitonin to permeabilize the plasma membrane and thapsigargin to inhibit SERCA, followed by treatment with FCCP to release all free-$Ca^{2+}$ from the mitochondrial matrix[20]. Quantification of $_mCa^{2+}$ content found that NCLX expression completely corrected APPswe-mediated $Ca^{2+}$ overload (Fig. 1m, n). In summary, AD-like stress is associated with a severe impairment in $_mCa^{2+}$ efflux capacity and predisposition to $_mCa^{2+}$-overload and permeability transition.

**Loss of neuronal NCLX accelerates AD progression.** To define whether impaired $_mCa^{2+}$ efflux causally contributes to the progression of AD, NCLX conditional mutant mice ($Slc8b1^{fl/fl}$, denoted as $NCLX^{fl/fl}$)[22] were crossed with a neuronal-restricted Cre recombinase transgenic model (Camk2a-Cre) to delete NCLX from the forebrain, specifically the prefrontal cortex and CA1 pyramidal cell layer in the hippocampus[33]. The resultant neuronal-specific NCLX conditional knockouts ($NCLX^{fl/fl}$ × Camk2a-Cre, NCLX-cKO) were backcrossed into the 3xTg-AD mutant mouse background (NCLX-cKO mice × 3xTg-AD, Fig. 2a). qPCR analysis of NCLX mRNA expression found ~75% loss of NCLX in the frontal cortex of 2-month-old NCLX-cKO × 3xTg-AD mice (Fig. 2b) and a corresponding reduction in hippocampal NCLX protein expression in 2, 9, and 12-month-old mice (Fig. 2c; Supplementary Fig. 7N–P). No changes in the expression of other proposed $_mCa^{2+}$ regulators were observed (Supplementary Fig. 2A–C and Supplementary Fig. 7Mm'–Xx'). Spatial learning and memory, was examined at 2, 6, 9, and 12 months of age using the Y-maze spontaneous alteration test[34]. Loss of neuronal NCLX greatly accelerated 3xTg-AD-associated impairments in spatial working memory as early as 6 months, which continued out to 12 months of age (Fig. 2d; Supplementary Fig. 2D). The total number of arm entries was not different between groups, suggesting these cognitive impairments were not owing to decreased fitness or motor function (Fig. 2e; Supplementary Fig. 2E). To further examine learning memory, we performed an automated contextual and cued fear-conditioning test commonly used to evaluate fear learning and memory[34,35]. Freezing behavior (complete immobility in this assay), is a common response to fearful situations. After animals experience pairing of an auditory cue with an electric footshock, the fear-producing stimulus results in freezing behavior, assessed as an index of associative fear learning and memory[36]. Freezing during the training session was equivalent among all groups, suggesting normal motor function (Fig. 2f; Supplementary Fig. 2F). Progressive impairments in contextual recall were noted in the 3xTg-AD mice from 9 to 12 months of age and deletion of neuronal NCLX potentiated this impairment in 12-month-old 3xTg-AD mice (Fig. 2g; Supplementary Fig. 2G). Impairments in cued recall were more substantial in NCLX-cKO × 3xTg-AD mice, which displayed a 50% reduction at 6 months of age, well before any decline was noted in 3xTg-AD mice. Cued recall continued to decline at an accelerated rate in NCLX-null mice out to 12 months of age (Fig. 2h; Supplementary Fig. 2H). This is intriguing as a deficit in cued recall is becoming clinically accepted as a sensitive indicator and predictor of dementia[37].

Intense research effort has been placed on identifying the link between $Ca^{2+}$ dysregulation and the amyloidogenic pathway[38].

Studies have suggested A$\beta$ increases $_iCa^{2+}$ levels by numerous mechanisms and that vice versa[39], increased $_iCa^{2+}$ augments A$\beta$ production and tau hyperphosphorylation, two hallmarks of AD. To examine the effect of neuronal genetic loss of $_mCa^{2+}$efflux on A$\beta$ formation in vivo, we measured the concentrations of soluble and insoluble A$\beta_{1-40}$ and A$\beta_{1-42}$ peptides in frontal cortex homogenates by ELISA. NCLX-cKO × 3xTg-AD mice brains displayed a significant increase in radio immunoprecipitation assay (RIPA) buffer -soluble A$\beta_{1-40}$ (~80%), A$\beta_{1-42}$ (~60%), and the A$\beta_{1-42}/_{1-40}$ ratio, as well as enhanced levels of insoluble A$\beta_{1-40}$ (~75%) and A$\beta_{1-42}$ (~85%) (Fig. 2i, j; Supplementary Fig. 2I). Immunohistochemistry identified amyloid deposits throughout the cerebral cortex and hippocampus of 3xTg-AD mice at 12 months of age and loss of neuronal NCLX increased the amyloid plaque burden by ~60% (Fig. 2k, l). To assess A$\beta$ production and processing we checked the expression of A$\beta$ precursor protein (APP), and various proteases involved in APP metabolism. We did not observe any changes in prefrontal cortex expression of total APP, $\alpha$-secretase (ADAM-10), or $\gamma$-secretase complex components (presenilin 1, PS1; APH1 subunit; and nicastrin) between 3xTg-AD brains and NCLX-cKO × 3xTg-AD samples. However, we did observe a significant increase in $\beta$-secretase (BACE1) expression with loss of NCLX expression in the 3xTg-AD background (Fig. 2m; Supplementary Fig. 7S). Beta-secretase is the rate-limiting enzyme in A$\beta$ production[40] and increased levels and activity of BACE1 protein are found in the brains of sporadic[41] and familial AD patients[42].

We next examined tau pathology, which is a pathological hallmark of AD and prominent in our mutant AD mouse model. Hyperphosphorylated tau forms insoluble aggregates that promote NFTs. The expression of total tau (soluble vs. insoluble) and tau phosphorylation at several epitopes in prefrontal cortex samples was evaluated by western blot and immunohistochemistry. Although phosphorylation was found to be elevated at all the tau residues examined in 3xTg-AD brains, we only found differential phosphorylation with the AT8 antibody (tau S202/T205) in addition to a significant increase in the total levels of insoluble tau in NCLX-cKO × 3xTg-AD brain homogenates (Fig. 2n; Supplementary Fig. 7X–C'). Consistent with the immunoblot results, HT7 total soluble-tau staining was not different between groups (Fig. 2o, p); while, AT8 (tau S202/T205) staining recapitulated the increase in phosphorylation observed by western blot (Fig. 2o, q). In summary, these results suggest that the loss of $_mCa^{2+}$ efflux capacity during AD progression accelerates impairments in memory and increases A$\beta$ plaque burden, tau hyperphosphorylation and histopathology.

**Rescue of neuronal NCLX expression prevents AD pathology.** To this point, our data suggest that $_mCa^{2+}$ overload may be a contributing factor to AD progression, so we next examined if enhancing $_mCa^{2+}$ efflux capacity would lessen disease burden. We generated a neuronal-specific, doxycycline-controlled, mutant mouse model to rescue NCLX expression, and function in the 3xTg-AD model (3xTg-AD × TRE-NCLX × Camk2a-tTA, Fig. 3a). To avoid developmental alterations in NCLX expression, mice were maintained on doxycycline until 4 weeks of age, after which it was removed to allow transgene activation. Camk2a-tTA-driven expression increased neuronal hippocampal NCLX mRNA and protein expression approximately twofold in 2 month old mice compared with age-matched controls (Fig. 3b, c; Supplementary Fig. 7D') with no change in the expression of mtCU components (Fig. 3c; Supplementary Fig. 7E'-G'). 3xTg-AD × NCLX-overexpressing mice and controls were evaluated for age-dependent loss of cognitive function and memory in the Y-maze and automated contextual and cued fear-conditioning tests.

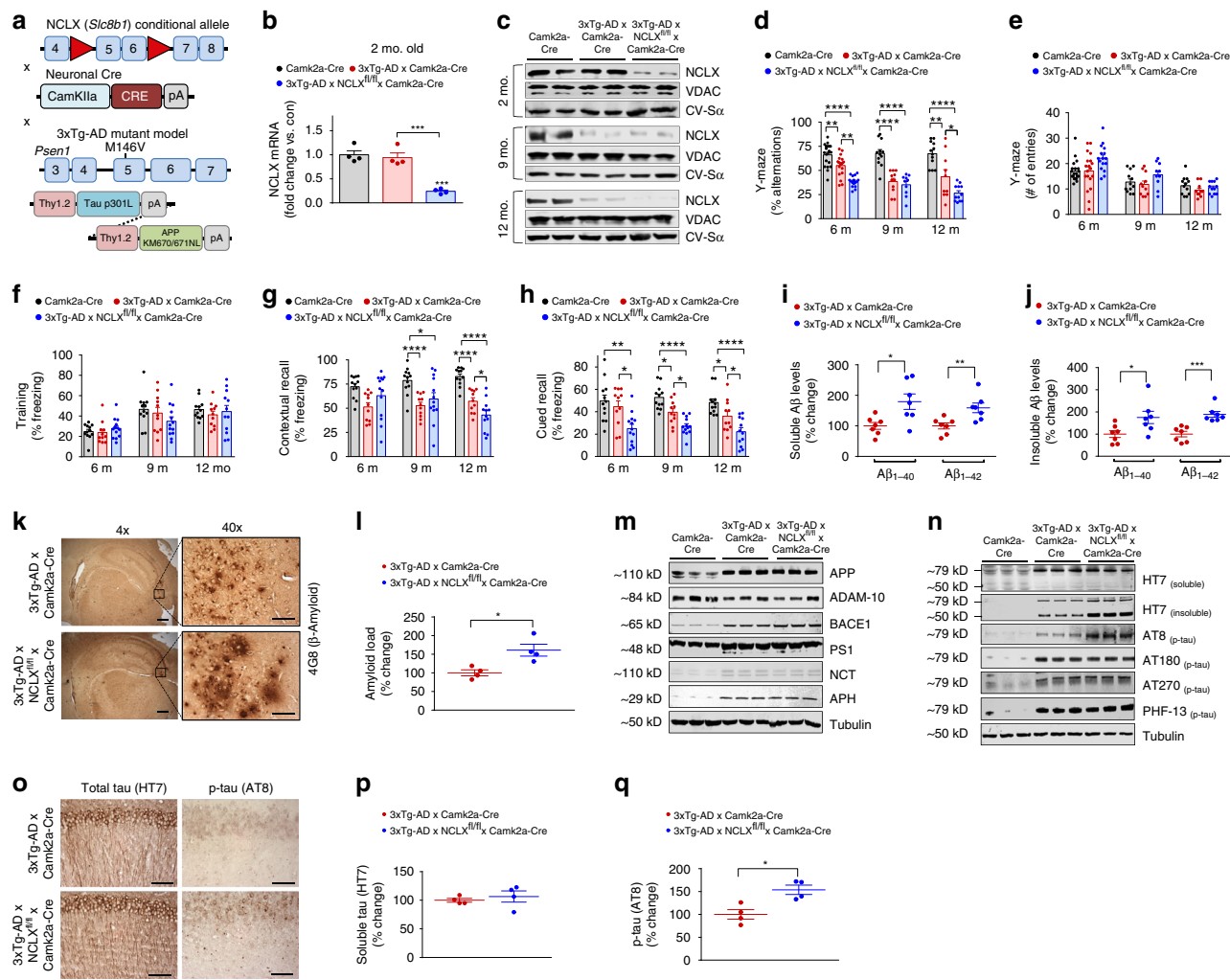

**Fig. 2** Neuronal deletion of NCLX accelerates AD pathology. **a** Schematic of NCLX knockout 3xTg-AD mutant mouse gene-targeting strategy. **b** NCLX mRNA expression, corrected to the housekeeping gene, Rps13; expressed as fold change vs. Camk2a-Cre control, $n = 4$ for all groups. **c** Western blots for NCLX expression in tissue isolated from the hippocampus of mice. VDAC and CV-Sα, served as mitochondrial loading controls. **d, e** Y-maze spontaneous alternation test. **d** Percentage of spontaneous alternation. **e** Total number of arm entries. **f–h** Fear-conditioning test. **f** Freezing responses in the training phase. **g** Contextual recall freezing responses, **h** Cued recall freezing responses. **i, j** Soluble and insoluble Aβ$_{1–40}$ and Aβ$_{1–42}$ levels in cortex of 12-month-old mice. **k** Representative immunohistochemical staining for 4G8-reactive β-amyloid; 4× scale bar = 100 μm, 40× scale bar = 50 μm. **l** Quantification of the integrated optical density area for Aβ immunoreactivity, $n = 4$ for all groups. **m** Western blots of full-length APP, ADAM-10, BACE1, PS1, Nicastrin, APH, and tubulin (loading control) for cortex homogenate of 12-month-old mice. **n** Representative western blots of soluble and insoluble total tau (HT7), phosphorylated tau at residues S202/T205 (AT8), T231/S235 (AT180), T181 (AT270), and S396 (PHF13) in cortex homogenate of 12-month-old mice, $n = 3$ for all groups. **o** Representative immunohistochemical staining for total tau (HT7) and phospho-tau S202/T205 (AT8) in hippocampus of mice; scale bar = 50 μm. **p, q** Quantification of the integrated optical density area of HT7 and AT8 immunoreactivity, $n = 4$ for all groups. ($n$ = individual dots shown for each group in all graphs. All data presented as mean ± SEM; ****$p < 0.001$, **$p < 0.01$, *$p < 0.05$; one-way ANOVA with Sidak's multiple comparisons test.) Source data are available as a Source Data file

Strikingly, the rescue of neuronal NCLX expression completely restored the age-associated cognitive decline of 3xTg-AD mice (Fig. 3d-h). In fact, 3xTg-AD × NCLX-overexpressing mice were no different from Camk2a-tTA controls in the number of alterations in the Y-maze test (spatial memory) and in freezing activity in relation to contextual and cued recall in the fear-conditioning test at all ages (6–12 months). Importantly, TRE-NCLX × Camk2a-tTA mice showed no baseline behavioral differences when compared to Camk2a-tTA controls (Supplementary Fig. 3A–E). These results suggest that the rescue of $_m$Ca$^{2+}$ efflux in the 3xTg-AD model is sufficient to suppress age-associated cognitive decline, even at an advanced stage of AD-like disease.

To evaluate if the protection against memory loss correlated with changes in overt cellular pathology, we evaluated Aβ levels in homogenates isolated from the frontal cortex. RIPA-soluble, and insoluble Aβ$_{1–40}$ levels were reduced by >50% in 3xTg-AD × NCLX-overexpressing mice and likewise Aβ$_{1–42}$ levels were significantly decreased, as compared with 3xTg-AD × Camk2a-tTA mice (Fig. 3i, j). In correlation, amyloid plaque burden was reduced (~50%) in the brains of 3xTg-AD × NCLX overexpressing mice (Fig. 3k, l). An evaluation of Aβ production and proteolytic processing revealed no difference in the expression of ADAM-10, or components of the γ-secretase complex (Fig. 3m; Supplementary Fig. 7H'–M'). However, we did observe a reduction in BACE1 expression in prefrontal cortex samples isolated from 3xTg-AD ×

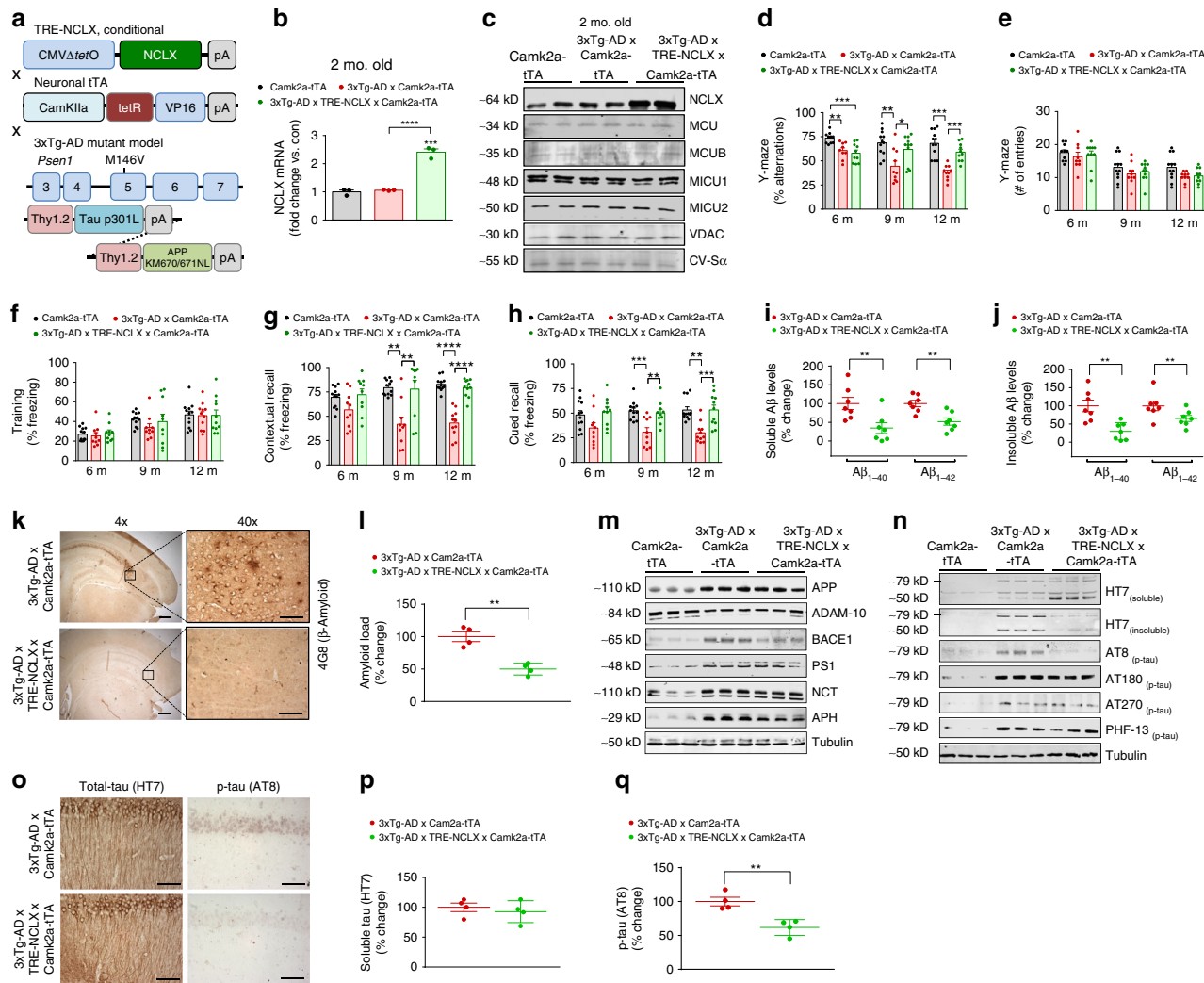

**Fig. 3** Neuronal rescue of NCLX expression impedes cognitive decline and AD pathology. **a** Schematic of tetracycline-responsive (TRE) transgenic construct and neuronal-specific driver, Camk2a-tTA. Neuronal-specific gain-of-function models were crossed with 3xTg-AD mutant mice to generate 3xTg-AD × TRE-NCLX × Camk2a-tTA mice. **b** NCLX mRNA expression corrected to the housekeeping gene, *Rps13*, expressed as fold change vs. tTA controls, $n = 3$ for all groups. **c** Western blots for NCLX expression and mitochondrial calcium uniporter channel components, tissue isolated from the hippocampus of 2 months old mice. **d, e** Y-maze spontaneous alternation test. **d** Percentage spontaneous alternation, **e** Total number of arm entries. **f–h** Fear-conditioning test. **f** Freezing responses in the training phase, **g** contextual recall freezing responses, **h** Cued recall freezing responses. **i, j** Soluble and insoluble $A\beta_{1-40}$ and $A\beta_{1-42}$ levels in brain cortex of 12 months old mice. **k** Representative immunohistochemical staining for 4G8-reactive β-amyloid; 4 × scale bar = 100 μm, 40 × scale bar = 50 μm. **l** Quantification of the integrated optical density area for Aβ immunoreactivity, $n = 4$ for all groups. **m** Western blots of full-length APP, ADAM-10, BACE1, PS1, Nicastrin, APH, and tubulin (loading control) in cortex homogenate of 12 months old mice, $n = 3$ for all groups. **n** Representative western blots of soluble and insoluble total tau (HT7), phosphorylated tau at residues S202/T205 (AT8), T231/S235 (AT180), T181 (AT270), and S396 (PHF13) in soluble brain cortex homogenate of 12 months old mice, $n = 3$ for all groups. **o** Representative hippocampal staining for total tau (HT7) and phospho-tau S202/T205 (AT8) immunoreactivity in 12 months old mice; scale bar = 50 μm. **p, q** Quantification of HT7 and AT8 integrated optical density area correct to 3xTg-AD × Camk2a-tTA controls, $n = 4$ for all groups. ($n$ = individual dots shown for each group in all graphs. All data presented as mean ± SEM; ****$p < 0.001$, **$p < 0.01$, *$p < 0.05$; one-way ANOVA with Sidak's multiple comparisons test.) Source data are available as a Source Data file

NCLX-overexpressing mice as compared with AD mutant mouse controls (Fig. 3m; Supplementary Fig. 7J'), a result that correlates with our findings of increased BACE1 expression in the NCLX-null AD model (Fig. 2m).

We next determined how the rescue of neuronal NCLX impacted tau pathology. There was a marked reduction (~40%) in total insoluble tau, as well as a significant decrease in phospho-tau at S202/T205 (AT8 immunoreactivity) (Fig. 3n; Supplementary Fig. 7P'-Q'). Phosphorylation of tau at Ser396 (PHF13), T181 (AT270), and T231/ S235 (AT180) residues were no different

between groups (Fig. 3n; Supplementary Fig. 7R'-T'). Immuno-histochemistry results further supported our biochemical findings, showing reduced levels (~50%) of phosphorylated tau at S202/T205 in the brains of 3xTg-AD mice overexpressing NCLX (Fig. 3o, q). We found no change in the somatodendritic labeling of total soluble-tau in CA1 pyramidal neurons, as would be expected in this model which features transgenic neuronal overexpression of mutant tau (Fig. 3o, p). In summary, the neuronal rescue of $_mCa^{2+}$ extrusion substantially reduced cognitive impairments and cytopathology in a robust in vivo model of AD.

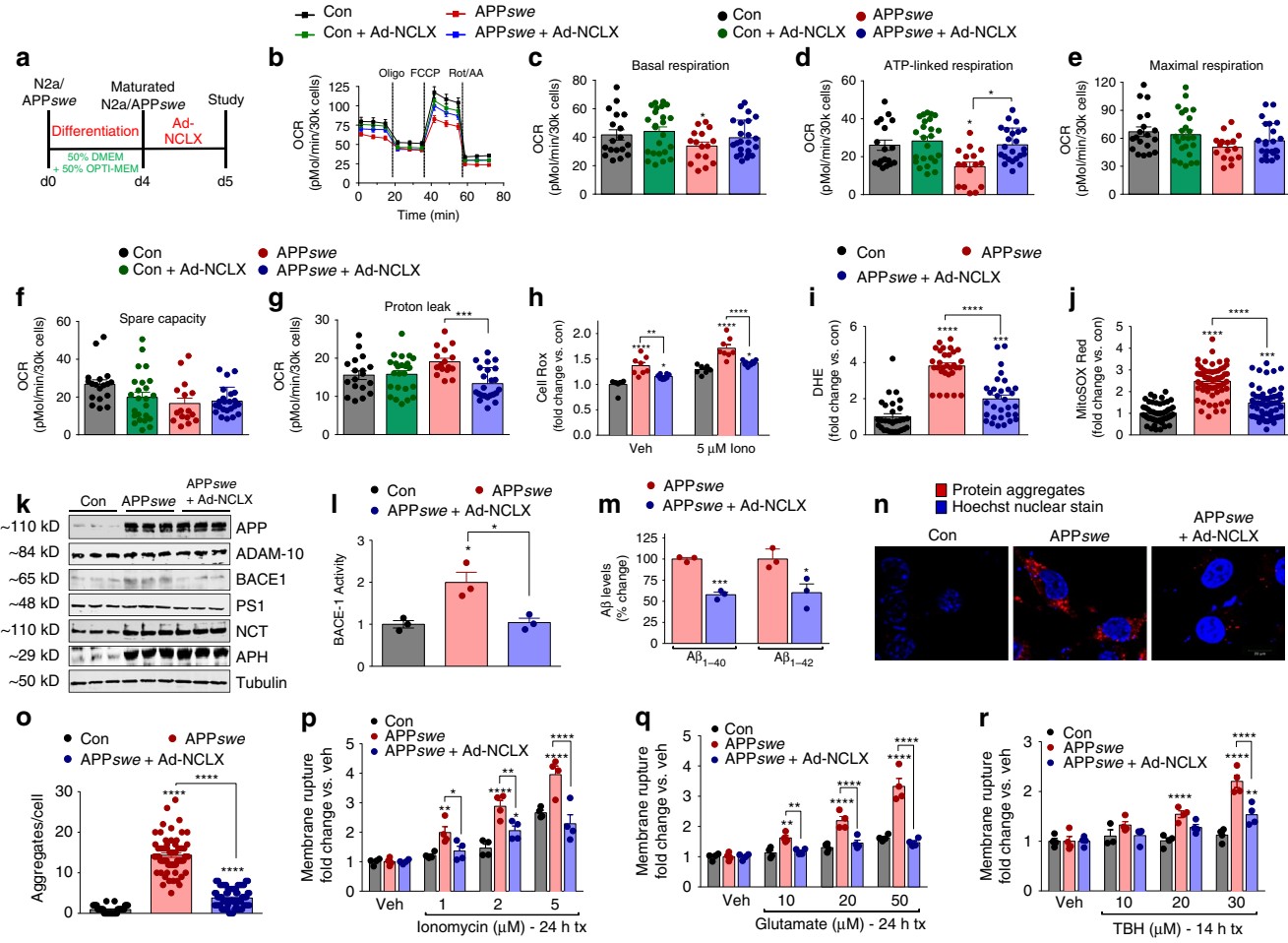

**Fig. 4** Enhancing $_mCa^{2+}$ efflux rescues mitochondrial dysfunction in APPswe cells. **a** Experimental protocol timeline for N2a maturation and adenovirus encoding NCLX (Ad-NCLX) transduction. **b** Oxygen consumption rate (OCR) at baseline and following: oligomycin (oligo; Complex V inhibitor; to uncover ATP-linked respiration), FCCP (protonophore to induce maximum respiration), and rotenone + antimycin A (Rot/AA; complex I and III inhibitor for complete ETC inhibition). **c** Quantification of basal respiration (base OCR – non-mito respiration (post-Rot/AA). **d** Quantification of ATP-linked respiration (post-oligo OCR−base OCR). **e** Maximum respiratory capacity (post-FCCP OCR−post-Rot/AA). **f** Spare respiratory capacity (post-FCCP OCR−basal OCR). **g** Proton leak (post-Oligo OCR−post-Rot/AA OCR). **h** Quantification of Cell Rox green fluorescent intensity (total cellular ROS production); fold change vs. N2a controls, $n = 8$ for all groups. **i** Quantification of DHE fluorescent intensity; fold change vs. N2a controls. **j** Quantification of MitoSOX fluorescent intensity; fold change vs. N2a controls, $n = 52$ for N2a control, $n = 59$ APPswe, and $n = 59$ for APPswe + Ad-NCLX. **k** Western blots of full-length APP, ADAM-10 (α-secretase) BACE1 (β-secretase), PS1, Nicastrin, APH (γ-secretase), and tubulin (loading con). **l** Quantification of β-secretase activity, $n = 3$ for all groups. **m** Quantification of extracellular $A\beta_{1-40}$ and $A\beta_{1-42}$ levels. **n** Representative images of intracellular protein aggregates in N2a control, APPswe and APPswe + Ad-NCLX cells stained with proteostat aggresome detection reagent (red) and Hoechst 33342 nuclear stain (blue), scale bars = 20 μm. **o** Total aggregates per cell, $n = 41$ for N2a control, $n = 62$ APPswe and $n = 69$ APPswe + Ad-NCLX. (P-R) Control, APPswe and APPswe transduced with Ad-NCLX for 48 h were assessed for plasma membrane rupture, Sytox Green, after treatment with: **p** Ionomycin ($Ca^{2+}$ ionophore, 1–5 μM), **q** glutamate (-NMDAR agonist, 10–50 μM), **r** tert-Butyl hydroperoxide (TBH, oxidizing agent, 10–30 μM), $n = 4$ experiments for each reagent. ($n$ = individual dots shown for each group in all graphs. All data presented as mean ± SEM; ****$p < 0.001$, **$p < 0.01$, *$p < 0.05$; one-way ANOVA with Sidak's multiple comparisons test.) Source data are available as a Source Data file

**Restoring NCLX expression improves mitochondrial function.** To define mechanistically how salvaging $_mCa^{2+}$ efflux and reducing $_mCa^{2+}$ overload diminishes AD-associated pathology we returned to our in vitro AD model and began probing for known $_mCa^{2+}$-linked cellular processes including oxidative phosphorylation (OxPhos), redox signaling, and cell death. Previously, metabolic regulation has been shown to have an important role in age-associated diseases, such as AD[43,44]. Maturated APPswe cells were monitored for changes in OxPhos by measuring mitochondrial oxygen consumption rates (OCR) in a Seahorse assay. The stable expression of mutant APPswe elicited a significant decrease in basal respiration, ATP-linked respiration,

and spare respiratory capacity with an increase in proton leak, all of which were rescued 48 h post NCLX expression (Fig. 4b–g). NCLX expression in control cells had no effect on OCR. Given the role of redox stress in AD pathology[45,46] and its close association with $_mCa^{2+}$ overload[47], coupled with the observation of increased ETC proton leak we examined the extent of oxidative stress. Thirty minutes following treatment with the $Ca^{2+}$ ionophore, ionomycin, APPswe displayed an increase in total ROS that was significantly reduced in APPswe cells expressing NCLX (48 h post adeno) (Fig. 4h). Next, we employed the superoxide probe, dihydroethidium (DHE), and found APPswe cells were generating approximately fourfold more superoxide, which was

significantly attenuated with the rescue of NCLX expression (Fig. 4i). To further define the subcellular source of ROS we measured superoxide production using a mito-targeted reporter, MitoSOX Red. Quantification of MitoSOX photometry revealed ~2.5-fold increase in mitochondrial superoxide generation in APP mutant cells and this was reduced by ~40% with NCLX expression (Fig. 4j).

Having established that the restoration of $_mCa^{2+}$ efflux capacity was sufficient to rescue OxPhos impairments and lessen oxidative stress, we next considered the impact on Aβ production, clearance, and toxicity. Western blot analysis in this model recapitulated our in vivo rescue of NCLX (see Fig. 3m). Although we found no change in total ADAM-10, or components of the γ-secretase complex, we again noted a significant increase in BACE1 expression in APPswe cells that was reduced with NCLX expression (Fig. 4k; Supplementary Fig. 6R'). In addition, BACE1 activity was increased twofold in APPswe cells and the rescue of NCLX expression reverted secretase activity back to control levels (Fig. 4l). To further evaluate the effect of NCLX expression on Aβ generation, we quantified extracellular $Aβ_{1-40}$ and $Aβ_{1-42}$ levels. Compared with APPswe neurons we observed a ~40% decrease in $Aβ_{1-40}$ and $Aβ_{1-42}$ formation in APPswe infected with Ad-NCLX (Fig. 4m). As Aβ oligomerization significantly contributes to aggregate formation we employed a fluorescent dye (ProteoStat) to quantify intracellular amyloid-like aggregates. APPswe cells displayed an abundance of intracellular aggregates (~14-fold over control), whereas the rescue of NCLX expression returned protein aggregation/inclusion body formation back to near control levels (Fig. 4n, o). These results are intriguing and suggest that elevated $_mCa^{2+}$ signaling/overload may contribute to the amyloid cascade. $_mCa^{2+}$-overload has been suggested to augment neuronal cell death, both through primary (MPTP and ROS) and secondary signaling mechanisms (metabolic derangement, etc.)[17,48]. NCLX expression was sufficient to reduce superoxide production and MPTP activation and enhance OxPhos capacity, so next we tested if these protective mechanisms coalesced to reduce neuronal demise. Control, APPswe, and APPswe cells + Ad-NCLX were treated with a variety of stressors associated with neurodegeneration including: ionomycin ($Ca^{2+}$ stress), glutamate (excitotoxicity), or tert-butyl hydroperoxide (TBH, ROS stress). Following treatment with these stressors plasma membrane rupture (hallmark of cell death) and cellular viability were quantified. The rescue of NCLX expression in AD cells significantly reduced cell death and increased viability across multiple doses of all three stressors tested (Fig. 4p, r and Supplementary Fig. 4A–C). These results support that the rescue of $_mCa^{2+}$ efflux in the context of AD may be a powerful therapeutic to impede cell loss and neurodegeneration.

**Rescue of $_mCa^{2+}$ efflux corrects mitochondrial dysfunction in AD.** To confirm our in vitro mechanistic findings, we measured the CRC of mitochondria isolated from the frontal cortex of 12 month old mice as an indicator of susceptibility to MPTP opening (Fig. 5a–d). 3xTg-AD mice (red arrow in representative tracings) showed a significant reduction in CRC compared with CamK2a-Cre controls (black arrow) (Fig. 5a, c). Accelerated loss of neuronal NCLX in 3xTg-AD mice (blue arrow) greatly decreased the CRC, suggesting 3xTg-AD × NCLX-cKO are more sensitive to MPTP activation and loss of membrane potential (Fig. 5a, b). Strikingly, the rescue of NCLX-dependent efflux significantly reduced MPTP opening in 3xTg-AD mice (Fig. 5c, d).

Given that $_mCa^{2+}$ overload is often associated with increased superoxide generation we performed DHE and 4-hydroxy-2-nonenal (4-HNE) staining to evaluate redox stress[46,49]. In agreement with our earlier findings the accelerated loss of

NCLX-dependent efflux in AD (3xTg-AD × NCLX-cKO) increased superoxide production in freshly sectioned cortex and hippocampal regions of the brain (Fig. 5e–g). In support, the restoration of $_mCa^{2+}$ efflux capacity by neuronal NCLX expression reduced AD-associated increases in superoxide production (Fig. 5h–j). Quantification of lipid peroxidation, oxidative degradation of lipids, by 4-HNE staining revealed a ~40% increase in 3xTg-AD × NCLX-cKO mice brains, as compared with 3xTg-AD × Camk2a-Cre mice (Fig. 5k, l). We noted a significant reduction in lipid peroxidation, indicative of reduced oxidative stress, in 3xTg-AD × NCLX over-expressing mice as compared with 3xTg-AD × Camk2a-tTA mice (Fig. 5m, n).

$_mCa^{2+}$-overload and mitochondrial dysfunction could lead to the activation of mitochondrial quality control pathways such as autophagy. To determine whether the protective effect of restoring NCLX expression in 3xTg-AD background was owing to the preservation of mitochondrial mass we quantified the ratio of mtDNA vs. nDNA, PGC1alpha mRNA expression, and citrate synthase activity[50]. Employing three different assays we found no change that would be indicative of a difference in mitochondrial content associated with the genetic modulation of neuronal NCLX (Fig. 5o, p; Supplementary Fig. 5A–D). We did observe a slight age-dependent decrease in mitochondrial content, but again this was independent of genotype.

## Discussion
Here we report that AD is associated with a loss in expression and functionality of the most prominent $_mCa^{2+}$ efflux transporter, NCLX, resulting in severe impairments in mitochondrial function and downstream signaling. The genetic rescue of NCLX expression in 3xTg-AD mice completely ablated age-associated cognitive decline and significantly reduced neuronal pathology. While previous reports have suggested a link between perturbed $_cCa^{2+}$ handling and mitochondrial dysfunction[6,8,9], to date no studies have directly examined the role of mitochondrial calcium signaling in AD or have addressed if it plays a casual role in disease progression. Postmortem AD brain pathology demonstrates increased oxidative stress and metabolic derangement, which is hypothesized to increase the vulnerability of neurons to excitotoxicity and cell death[51]. We and others have previously reported that impairments in $_mCa^{2+}$ exchange can significantly alter metabolism and cell death[20,22,52], both of which have been shown to contribute to neurodegeneration[13] through the activation of various cell death pathways[17,19]. This paradigm fits with our previous report[18] where we found that impairments in $_mCa^{2+}$ exchange contributed to the development and progression of heart failure, supporting that $_mCa^{2+}$ efflux is likely an important contributor to cellular homeostasis and function in the context of chronic disease.

There are several mechanisms by which $Ca^{2+}$ has been linked to neurodegenerative diseases. $_mCa^{2+}$ is known to cause OMM permeability provoking the release of apoptogens[53]. Further, $_mCa^{2+}$ is a central priming event in the opening of the mitochondrial permeability transition pore (MPTP) causing the collapse of membrane potential and loss of ATP production, resulting in necrotic cell death[20]. In support of the centrality of this pathogenic mechanism, the inhibition of MPTP activation using both pharmacological (cyclosporine-A and its derivatives) and genetic means (CypD KO) reduces neuronal dysfunction and degeneration in both cell culture and mutant mouse AD models[17,54]. In agreement with this premise, Logan et al. [55] reported that loss-of-function mutations in MICU1 (a negative regulator of mtCU at low-/homeostatic $_cCa^{2+}$ levels) promotes $_mCa^{2+}$ overload and is associated with severe brain and muscle

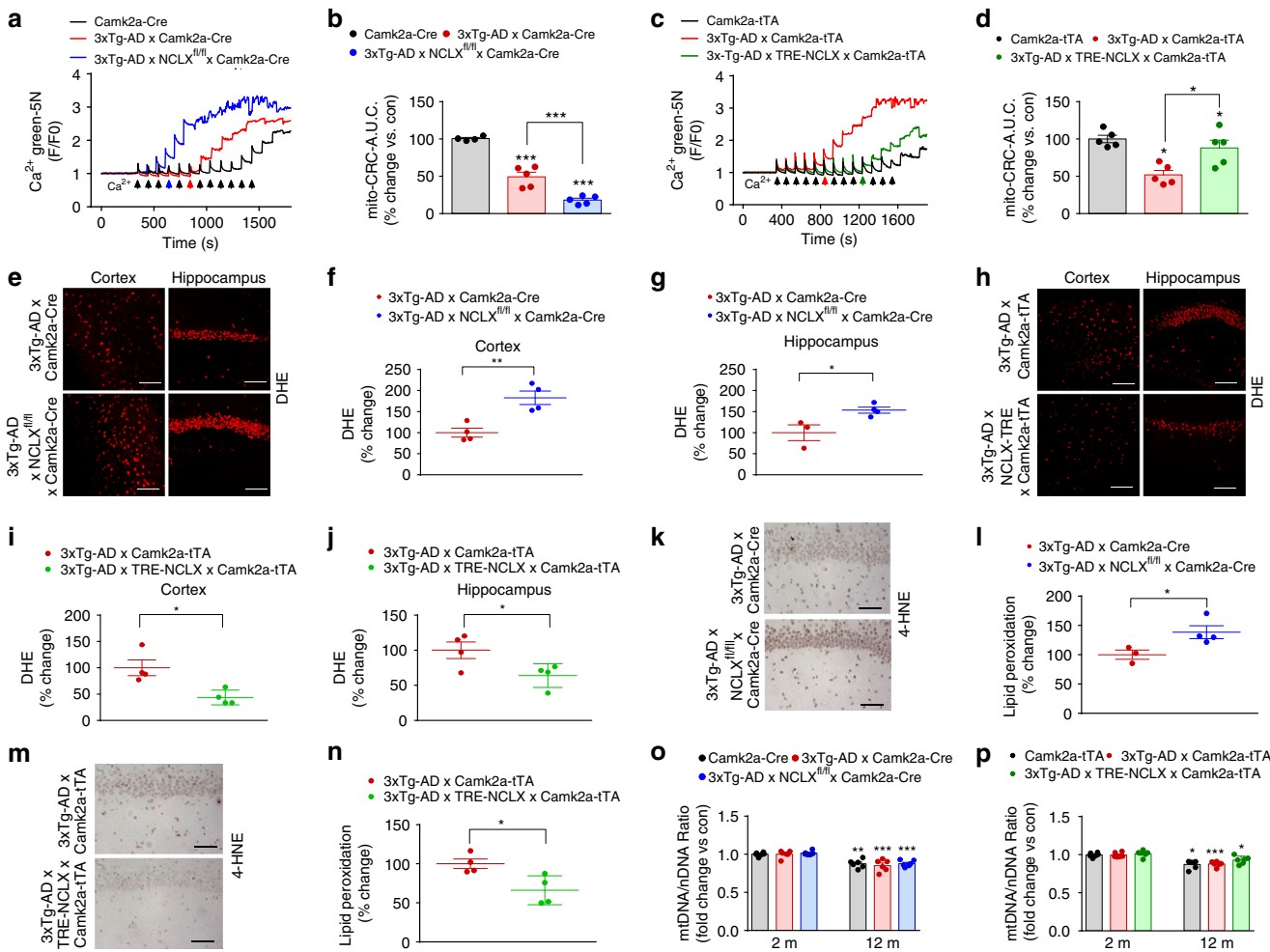

**Fig. 5** Restoration of $_mCa^{2+}$ efflux improves mitochondrial function in AD mutant mice. **a** Representative traces for $_mCa^{2+}$ retention capacity (CRC). **b** Percent change in CRC of 3xTg-AD × Camk2a-Cre and 3xTg-AD × NCLX$^{fl/fl}$ × Camk2a-Cre vs. Camk2a-Cre control. **c** Representative trace for CRC in Camk2a-tTA, 3xTg-AD × Camk2a-tTA and 3xTg-AD × TRE- NCLX × Camk2a-tTA mice. **d** Percent change in $_mCa^{2+}$ retention capacity of 3xTg-AD × Camk2a-tTA and 3xTg-AD × TRE- NCLX × Camk2a-tTA vs. Camk2a-tTA control. **e** DHE staining for ex vivo detection of superoxide production in freshly prepared cortical and hippocampal sections from 12 months old mice. **f**, **g** DHE fluorescent intensity, percent change vs. 3xTg-AD × Camk2a-Cre controls. **h** DHE staining for ex vivo detection of superoxide production in freshly prepared cortical and hippocampal sections from 12 months old mice. **i**, **j** Quantification of DHE fluorescent intensity, percent change vs. 3xTg-AD × Camk2a-tTA controls. **k** Representative images of hippocampal 4-HNE immunohistochemistry to detect lipid peroxidation in 12 months old mice. **l** Percent change in 4-HNE-integrated optical density area corrected to 3xTg-AD × Camk2a-Cre controls. **m** Representative images of hippocampal 4-HNE immunohistochemistry to detect lipid peroxidation in 12 months old mice. **n** Percent change in 4-HNE-integrated optical density area corrected with 3xTg-AD × Camk2a-tTA controls. **o** Mitochondrial DNA (mtDNA)/nuclear DNA (nDNA) ratio in tissue isolated from the cortex of 2 and 12 months old mice, fold change vs. 2 months old Camk2a-Cre controls. **p** mtDNA/nDNA ratio in tissue isolated from the brain cortex of 2 and 12 months old mice expressed as fold change vs. 2 months old Camk2a-tTA controls. ($n$ = individual dots shown for each group in all graphs. All data presented as mean ± SEM; ****$p < 0.001$, **$p < 0.01$, *$p < 0.05$; one-way ANOVA with Sidak's multiple comparisons test.) Source data are available as a Source Data file

disorders. These clinical results provide strong correlative evidence that alterations in $_mCa^{2+}$ are linked with human disease. In support of this notion, increased $_mCa^{2+}$ uptake via ERK1/2 dependent upregulation of MCU was recently reported to elicit dendritic injury in a late-onset familial Parkinson's disease (PD) model (mutation in Leucine-Rich Repeat Kinase 2)[56]. Further, the phosphorylation of NCLX by protein kinase A (PKA), which is suggested to augment its activity, has been shown to reduce $_mCa^{2+}$ load and enhance neuronal survival in a PINK-1 knock-down cell line[57]. Other studies highlight the relationship between $_mCa^{2+}$ dysregulation and alterations in ER and mitochondria contact sites, which may also contribute to PD[58] and AD pathogenesis[16,59]. This is intriguing, as others have suggested that

Aβ mutations increase ER-mitochondrial contact[16,59], which could result in elevated ER to mitochondria $Ca^{2+}$ transfer and promote matrix-overload. In total, these studies support our working hypothesis that $_mCa^{2+}$ homeostasis is disrupted in AD leading to cellular dysfunction promoting a vicious pathological cycle contributing to disease progression.

The observed $_mCa^{2+}$ dysregulation was associated with a significant increase in mitochondrial superoxide generation and impaired mitochondrial energetics. A number of studies have shown that oxidative stress precedes Aβ accumulation and tau phosphorylation[46,49] and is an early indicator of AD. Oxidative stress has also been shown to increase BACE1 expression, which was likewise observed here[60]. In addition to oxidative stress[61],

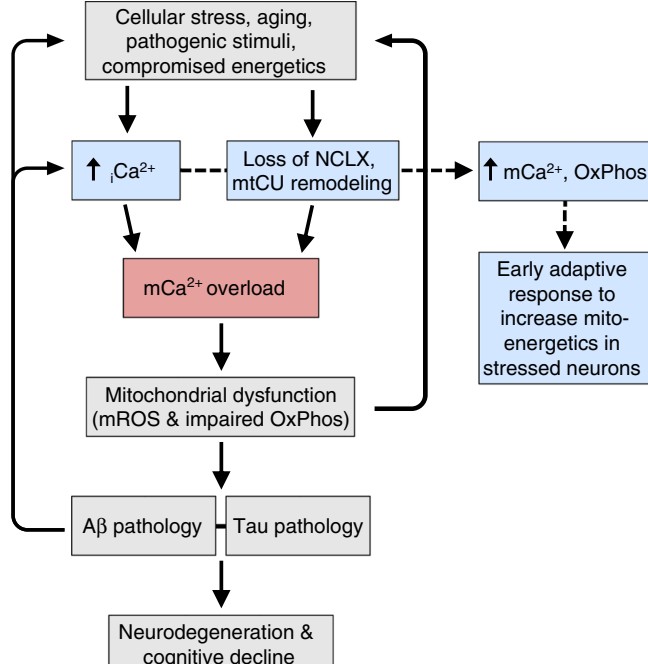

**Fig. 6** Working hypothesis of $_mCa^{2+}$ exchange dysfunction in AD pathogenesis. Alzheimer's disease (AD) initiators, such as aging and metabolic dysfunction, elicit a compensatory increase in intracellular calcium ($_iCa^{2+}$) and remodeling of the $_mCa^{2+}$ exchange machinery to elevate matrix $Ca^{2+}$ levels and activate mitochondrial dehydrogenases to augment cellular energetics (blue boxes). However, these alterations in $_iCa^{2+}$ handling quickly turn maladaptive leading to $_mCa^{2+}$-overload, which results in mitochondrial dysfunction and AD pathophysiology. The proposed sequence of events initiates positive feedback at multiple levels of the disease pathway potentiating the progression of neurodegeneration

increased levels of BACE1 expression and tau hyperphosphorylation have been reported during energy depletion[62], and mitochondrial stress conditions[63]. Therefore, it is intriguing to surmise that $_mCa^{2+}$ overload and the associated oxidative stress may form a maladaptive feed-forward loop with BACE1-mediated amyloidosis. If correct, we may have uncovered another target to inhibit the Aβ cascade, which is the predominant mechanism thought to reduce synaptic dysfunction in AD[64,65].

A few outstanding questions remain, such as why $_mCa^{2+}$ efflux is downregulated early in the pathogenesis of AD. Our current working hypothesis is that $_mCa^{2+}$ efflux is decreased and uptake is increased in an attempt to elevate matrix $Ca^{2+}$ content and augment metabolic signaling, such as dehydrogenase activity, and ATP production. This early compensatory response in the metabolically challenged/stressed neuron then quickly turns maladaptive due to $_mCa^{2+}$ overload and the sequella we report here (Fig. 6). In addition, we were surprised that genetic alterations in NCLX had no discernable impact on basal neuronal function or behavior in young 2 months old mice given the reported role of mitochondria to buffer $Ca^{2+}$ at the synapse[66,67]. We believe that perhaps the level of buffering required for synaptic transmission remained intact in our model since we did not alter MCU-mediated uptake and only enhanced efflux capacity. While much work remains to unravel many of the downstream mechanisms contributing to neurodegeneration in AD, including the role of mtCU remodeling and $_mCa^{2+}$ uptake, our study provides the first conclusive biological evidence that loss of $_mCa^{2+}$ efflux accelerates AD development and that enhancing the clearance of pathogenic $_mCa^{2+}$ may represent a powerful therapy to reduce the progression of AD and other neurodegenerative diseases.

## Methods

**Neuronal-specific NCLX knockout 3xTg-AD mutant mouse**. NCLX floxed mice were generated by our lab[22] by acquiring targeted ES cells generated by recombinant insertion of a knockout-1st mutant construct containing loxP sites flanking exons 5–7 of NCLX, *Slc8b1* gene (ch12: 113298759- 113359493)[68]. ES cell lines (clone EPD0460_4_A08, EUCOMM) were confirmed by PCR and injected into C57BL/6N blastocysts with subsequent transplantation into pseudo- pregnant females. Germline mutant mice were crossed with ROSA26-FLPe knock-in mice for removal of the FRT-flanked splice acceptor site, βgal reporter, and neomycin resistance cassette. Resultant NCLX$^{fl/+}$ mice were interbred to generate homozygous mutant mice with knockout potential (NCLX$^{fl/fl}$). Homozygous LoxP 'floxed' mice (NCLX$^{fl/fl}$) were crossed with neuron-specific Camk2a-Cre recombinase driver lines (available from Jackson Laboratory, stock no. 005359), resulting in germline neuronal-specific deletion of NCLX. The Calcium/calmodulin-dependent protein kinase II alpha (Camk2a) promoter drives Cre recombinase expression in the forebrain, specifically to the CA1 pyramidal cell layer in the hippocampus[33]. These mice were viable and fertile. Resultant neuronal-specific loss-of-function models (NCLX-cKO- NCLX$^{fl/fl}$ Camk2a-Cre) were crossed with 3xTg-AD mutant mouse[24] (available from Jackson Laboratory, stock no. 34830), to generate 3xTg-AD × NCLX$^{fl/fl}$ × Camk2a-Cre (3xTg-AD × NCLX-cKO mice) mutant mice. 3xTg-AD mice are homozygous for the Psen1 mutation (M146V knock-in), and contain transgenes inserted into the same loci expressing the APP*swe* mutation (APP KM670/671NL) and tau mutation (MAPT P301L). All experiments involving animals were approved by Temple University's IACUC and followed AAALAC guidelines. All mice were maintained in our animal facility under pathogen-free conditions on a 12 h light/12 h dark cycle with continuous access to food and water.

**Neuronal-specific NCLX overexpression 3xTg-AD mutant mouse**. We cloned the human NCLX sequence (NM_024959) (5′ EcoRI, 3′ XmaI) into a plasmid containing the Ptight Tet-responsive promoter and a SV40 poly(A) sequence and linearized the construct with XhoI digestion followed by gel and Elutip DNA purification. Upon sequence confirmation the purified fragment was injected into the pronucleus of a fertilized ovum and transplanted into pseudo-pregnant females (C57BL/6N)[22]. Upon confirmation of germline transmission in founder lines, mutant mice were crossbred with the Camk2a-tTA (neuronal-restricted expression, doxycycline-off; Jackson Laboratory, stock no. 003010) transgenic model[69]. This allowed conditional overexpression upon the withdrawal of chow containing doxycycline (a tetracycline analog). Resultant neuronal-specific gain-of-function model (TRE-NCLX × Camk2a-tTA) were crossed with 3xTg-AD mutant mouse to generate 3xTg-AD × TRE-NCLX × Camk2a-tTA mutant mice. For all experiments, mice were approved by Temple University's IACUC following all AAALAC guidelines.

**Human AD tissue samples**. Frontal cortex samples were collected postmortem from non-familial AD patients and age-matched controls with no history of dementia. These samples were provided by Dr. Domenico Praticò via the Arizona State University (ASU) Brain bank. All tissue samples were rapidly frozen in liquid nitrogen and stored at −80 °C until isolation of protein (n = 7 for non-familial, sporadic AD and n = 7 for age-matched no history of dementia controls). Patient consent, sample collection and preparation, and clinical data collection were performed per a protocol approved by Temple University School of Medicine Institutional Review Board. Demographics of human samples are given in Supplementary Table 1.

**Cell cultures and differentiation**. Mouse neuroblastoma N2a cell line as control cells (N2a/con) and N2a cells stably expressing human APP carrying the K670N, M671L Swedish mutation (APP*swe*) were grown in Dulbecco's modified Eagle's medium supplemented with 10% fetal bovine serum, 1% penicillin/streptomycin (GIBCO) and in the absence (N2a/con) or presence of 400 μg/mL G418 (APP*swe*) (Invitrogen Corporation) at 37 °C in the presence of 5% CO2. Before experiments, APPswe cells were maintained to regular media without G418. In differentiation studies, cells were grown in 50% Dulbecco's modified Eagle's medium (DMEM), 50% OPTI-MEM, 1% penicillin/streptomycin (GIBCO) for 72 hrs. Only cells with passage number <20 were used. For all imaging studies, cells were plated on glass coverslips pre-coated with poly-ᴅ-lysine. For overexpression of NCLX, we infected N2a Con and APP*swe* cells with adenovirus encoding NCLX (Ad-NCLX) for 48 hrs.

**qPCR mRNA analysis**. RNA was extracted using the Qiagen RNeasy Kit. In brief, 1 μg of total RNA was used to synthesize cDNA in a 20 μL reaction using the High-Capacity cDNA Reverse Transcription Kit (Applied Biosystems). qPCR analysis was conducted following manufacturer instructions (Maxima SYBR, Thermo Scientific). RPS13 was always used as an internal control gene to normalize for RNA. Each sample was run in duplicate, and analysis of relative gene expression was

calculated using the 2^ΔΔCt method. A list of all primer sequences can be found in Supplementary Table 2.

**Live-cell imaging of Ca²⁺ transients**. Matured neuronal cells were infected with AAV6-$_{mito}$R-GECO to measure $_m$Ca²⁺ dynamics or loaded with the cytosolic Ca²⁺ indicator, 5 μM Fluo4-AM to study cytosolic Ca²⁺ dynamics. Cells were imaged continuously in Tyrode's buffer (150 mM NaCl, 5.4 mM KCl, 5 mM HEPES, 10 mM glucose, 2 mM CaCl₂, 2 mM sodium pyruvate at pH 7.4) on a Zeiss 510 confocal microscope. Cells were treated with the depolarizing agent, 100 mM KCl, to activate voltage-gated calcium channels during continuous live-cell imaging. $_m$Ca²⁺ efflux rate was calculated as $(F_{max}/F_0)-(F_{end}/F_0)/sec$ and units expressed as ΔF/sec and $_m$Ca²⁺ peak amplitude was calculated as $(F_{max}/F_0)-(F_0/F_0)$. ΔF = change in fluorescence intensity, $F_0$ = fluorescence intensity at 0 sec, $F_{max}$ = maximum fluorescence intensity, $F_{end}$ = fluorescence intensity at the endpoint. Time was presented in seconds.

**Evaluation of $_m$Ca²⁺ retention capacity and content**. To evaluate $_m$Ca²⁺ retention capacity (CRC) and content, N2a as con, APPswe and APPswe infected with Ad-NCLX for 48 h were transferred to an intracellular-like medium containing (120 mM KCl, 10 mM NaCl, 1 mM KH2PO4, 20 mM HEPES-Tris), 3 μM thapsigargin to inhibit SERCA so that the movement of Ca²⁺ was only influenced by mitochondrial uptake, 80-μg/ml digitonin, protease inhibitors (Sigma EGTA-Free Cocktail), supplemented with 10 μM succinate and pH to 7.2. All solutions were cleared with Chelex 100 to remove trace Ca²⁺ (Sigma). For $_m$Ca²⁺ retention capacity: $2 × 10^6$ digitonin-permeabilized neuronal cells were loaded with the ratiometric reporters FuraFF at concentration of 1 μM (Ca²⁺). At 20 s JC1 (Enzo Life Sciences) was added to monitor mitochondrial membrane potential (Δψ). Fluorescent signals were monitored in a spectrofluorometer (Delta RAM, Photon Technology Int.) at 340- and 380-nm ex/510-nm em. After acquiring baseline recordings, at 400 s, a repetitive series of Ca²⁺ boluses (10 μM) were added at the indicated time points. At completion of the experiment a protonophore, 10 μM FCCP, was added to uncouple the Δψm and release matrix free-Ca²⁺. All experiments (three replicates) were conducted at 37 °C. For $_m$Ca²⁺ content cells from all the groups were loaded with Fura2 and treated with digitonin and thapsigargin[20]. Upon reaching a steady state recording, a protonophore, FCCP, was used to collapse ΔΨ and initiate the release of all matrix free-Ca²⁺.

To examine $_m$Ca²⁺ retention capacity in vivo, mitochondria was isolated from frontal cortex of 12 months old mice in mitochondria isolation buffer (225 mM Mannitol, 75 mM Sucrose, 5 mM MOPS, 0.5 mM, 2 mM Taurine). Isolated mitochondria were transferred to an intracellular-like medium containing (120 mM KCl, 10 mM NaCl, 1 mM KH2PO4, 20 mM HEPES-Tris) supplemented with 10 μM succinate. In total, 200 μg mitochondria were loaded with the 1 μM Calcium Green 5 N to monitor extra-mitochondrial Ca²⁺. Repetitive Ca²⁺ boluses (5 μM) were added after baseline recordings at 400 s. Fluorescent signal at ($488_{Ex}/530_{Em}$) was measured using plate reader. Inverse area under the curve was calculated to measure percentage CRC after stepwise addition of 50 μm Ca²⁺ boluses.

**Western blot analysis**. All protein samples from brain or cell lysates were lysed by homogenization in RIPA buffer for the soluble fractions and then in formic acid (FA) for the insoluble fractions and used for western blot analyses. Samples were run by electrophoresis on polyacrylamide Tris-glycine sodium dodecyl sulphate gels. The following primary antibodies were used in the study: MCU dilution 1:1000 (Sigma-Aldrich, Catalog # HPA016480), MCUB dilution 1:1000 (Abgent, Catalog #AP12355b), MICU1 dilution 1:500 (Custom generation by Yenzyme), MICU2 dilution 1:1000 (Abcam, Catalog # ab101465), EMRE dilution 1:1000 (Santa Cruz, Catalog # sc-86337), NCLX dilution 1:500 (NCKX6 Santa Cruz, Catalog # sc-161921), NCLX dilution 1:250 (Abcam SLC24A6, Catalog # ab83551), VDAC dilution 1:2500 (Abcam, Catalog # ab15895), ETC respiratory chain complexes dilution 1:5000 (OxPhos Cocktail, Abcam, Catalog # MS604), anti-APP N-terminal raised against amino acids 66–81 for total APP 22C11 dilution 1:1500 (Chemicon International, Catalog # MAB348), BACE1 dilution 1:500 (Sigma, Catalog # MAB5308), ADAM-10 dilution 1:500 (Chemicon International, Catalog # AB19026), PS1 dilution 1:500 (Sigma- Aldrich, Catalog # S182), nicastrin dilution 1:200 (Cell Signaling, Catalog # 5665), APH1 dilution 1:200 (Millipore, Billerica, MA, Catalog # AB9214), total tau HT7 dilution 1:200 (Thermo Fisher Scientific, Catalog # MN1000), phospho-tau (pThr231) Monoclonal Antibody (AT180) dilution 1:200 (Thermo Fisher Scientific, Catalog # MN1040), phospho-Tau (Ser202, Thr205) monoclonal antibody (AT8) dilution 1:200 (Thermo Fisher Scientific, Catalog # MN1020), phospho-tau (pThr181) antibody (AT270) dilution 1:200 (Thermo Fisher Scientific, Catalog # MN1050); phospho-tau (pS396) antibody (PHF13) dilution 1:200 (Cell Signaling, Catalog # 9632), beta-Tubulin dilution 1:1000 (Abcam Catalog # ab6046) and Licor IR secondary antibodies dilutions 1:10,000 (LI-COR Biosciences, anti-mouse, Catalog # 925–68070; anti-rabbit, Catalog # 926–32211; anti-goat, Catalog # 926–32214). All blots were imaged on a Licor Odyssey system. All full-length western blots are available in Supplementary Fig. 6. Densitometry of all the western blots are shown in Fig. S7.

**Cognition function tests**. Mice at 2, 6, 9, and 12 months of age were assessed for behavioral test in the Y-maze and fear-conditioning assay.

**Y-maze**: In this test, mice were placed in the center of the Y-maze, and allowed to explore freely through the maze during a 5-min session. This apparatus consisted of three arms 32 cm (long) 610 cm (wide) with 26-cm walls. The sequence and total number of arms entered were recorded. An entry into an arm was considered valid if all four limbs entered the arm. An alternation was defined as three consecutive entries in three different arms (i.e., 1, 2, 3, or 2, 3, 1). The percentage alternation score was calculated using the following formula: total alternation number/total number of entries-2) × 100. Furthermore, total number of arm entries was used as a measure of general activity in the animals. The maze was wiped clean with 70% ethanol between each animal to minimize odor cues.

**Fear conditioning**: In brief, the fear-conditioning test was performed in a fear-conditioning chamber equipped with black methacrylate walls, a transparent front door, a speaker, and grid floor (Start Fear Systems; Harvard Apparatus). During the training phase, each mouse was placed in the chamber and underwent three cycles of 30 s of sound and 10 s of electric shock within a 6-minute time interval. The next day, the mouse spent 5 min in the chamber without receiving electric shock or hearing the sound (contextual recall). Two hours later, the animal spent 6 min in the same chamber but with different flooring, walls, smells, and lighting, and heard the cued sound for 30 s (cued recall). Freezing activity of the mouse was recorded for each phase. The formation of context fear is associated with the hippocampus and recall of associations to cues is linked with the amygdala, therefore, we can assess hippocampus and amygdala associated memory using this test. In this assay, if the mouse remembers and connects the environment with the stimulus, it will freeze, and freezing response is measured as a read-out.

**Immunohistochemistry**. Mouse brains were prepared for immunohistochemistry. Serial 6-μm thick sections were deparaffinized, hydrated, and blocked in 2% fetal bovine serum before incubation with primary antibody overnight at 4 °C. Sections were incubated overnight at 4 °C with primary antibodies Aβ-4G8 (1:150 Covance), HT7 (1:150), AT8 (1:50), 4-HNE (1:20; ab48506,) then incubated with secondary antibody and developed using the avidin–biotin complex method (Vector Laboratories, Burlingame, CA) with 3,3′diaminobenzidine as chromogen. 4G8 antibody is reactive to amino-acid residues 17–24 of Aβ and the epitope lies within amino acids 18–22 of Aβ. 4G8 Aβ antibody reacts to abnormally processed iso-forms, as well as precursors. The epitope for HT7 lies within amino acids 159–163 of tau.

**Biochemical analysis**. Mouse brain homogenates were sequentially extracted first in RIPA for the soluble fractions and then in FA for the insoluble fractions. In brief, 30 mg of cerebral cortex was sonicated in RIPA buffer added with protease and phosphatase inhibitors cocktail and subsequently ultracentrifuged at 90,720×g for 45 min. Supernatants were used to measure Aβ and tau soluble fractions by enzyme-linked immunosorbent assay (ELISA) and western blotting, respectively. Pellets were mixed in 70% formic acid, sonicated, neutralized in 6 N sodium hydroxide, and used to measure Aβ and tau insoluble fractions by ELISA and western. Aβ$_{1–40}$ and Aβ$_{1–42}$ levels were assayed by ELISA kit (Wako Chemicals USA, Inc.). For in vitro analysis of Aβ$_{1–40}$ and Aβ$_{1–42}$ levels, conditioned media from APPswe cells and cells infected Ad-NCLX were collected and analyzed at a 1:100 dilution using the method described in the manual (Wako Chemicals USA, Inc.) as described previously[70]. Aβ$_{1–40}$ and Aβ$_{1–42}$ in samples were captured with the monoclonal antibody BAN50, which specifically detects the N-terminal of human Aβ$_{(1–16)}$. Captured human Aβ is recognized by another antibody, BA27 F (Aβ')2-HRP, a mAβ specifically detects the C-terminal of Aβ$_{40}$, or BC05 F(Aβ')2-HRP, a mAβ specific for the C-terminal of Aβ$_{42}$, respectively. HRP activity was assayed by color development using TMB. The absorbance was then measured at 450 nm. Values were reported as a percentage of Aβ$_{1–40}$ and Aβ$_{1–42}$ secreted relative to control.

**Evaluation of ROS production**. To measure the total cellular ROS, we employed CellROX Green, a cell-permeable non-fluorescent (or very weakly fluorescent in a reduced state) fluorogenic probes that exhibits a strong fluorogenic signal upon oxidation. In this assay, cells were loaded with CellROX Green Reagent at a final concentration of 5 μM for 30 min at 37 °C and measured the fluorescence at 485/ex and 520/em using a Tecan Infinite M1000 Pro plate reader. To measure mito-chondrial superoxide production cells were loaded with 10 μM MitoSOX Red for 45 min at 37 °C and imaged at 490/20ex and 585/40em. To monitor O₂ generation, cells from all groups was stained with 20 μM DHE for 30 min at 37 °C and imaged on Carl Zeiss 510 confocal microscope at 490/20ex and 632/60em. For in vivo detection of superoxide levels, cortical, and hippocampal slices were freshly prepared from mice brains and stained with 20 μM DHE for 30 min at 37 °C and imaged on a Carl Zeiss 710 confocal microscope. Images were quantified for fluorescent optical density using ImageJ.

**Mitochondrial bioenergetics**. Control (N2a), APPswe, and Con, APPswe cells infected with Ad-NCLX for 48 h were subjected to OCR measurement at 37 °C in an XF96 extracellular flux analyzer (Seahorse Bioscience). Cells ($3 × 10^4$) were plated in XF media pH 7.4 supplemented with 25 mM glucose and 1 mM sodium

pyruvate and sequentially exposed to oligomycin (1.5 μM), FCCP(1 μM), and rotenone plus antimycin A (0.5 μM). Quantification of basal respiration (base OCR – non-mito respiration (post-Rot/AA), ATP-linked respiration (post-oligo OCR −base OCR), Max respiratory capacity (post-FCCP OCR−post-Rot/AA), Spare respiratory capacity (post-FCCP OCR−basal OCR) and Proton leak (post-Oligo OCR−post-Rot/AA OCR) was performed.

**Membrane rupture and cell viability assay**. We evaluated membrane rupture using SYTOX Green (Life Technologies) a membrane impermeable fluorescent stain, which enters the cell upon membrane rupture, intercalates in DNA and increases fluorescence > 500-fold. We examined general cell viability using Cell Titer Blue (resazurin, Life Technologies). This Cell Titer Blue assay uses the indicator dye resazurin to measure the metabolic capacity of cells. Viable cells retain the ability to reduce resazurin into resorufin, which is highly fluorescent. Nonviable cells rapidly lose metabolic capacity, do not reduce the indicator dye, and thus do not generate a fluorescent signal. Equal numbers of N2a, APP*swe* and APP*swe* infected with Ad-NCLX for 48 h were treated with Ionomycin, (1–5 μM) for 24 h and an oxidizing agent TBH (10–30 μM) for 14 h and glutamate (NMDAR-agonist, neuroexcitotoxicity agent) (10–50 μM) for 24 h. On the day of the experiment, cells were loaded with 1 μM Sytox green for 15 min at 37 °C and measured the fluorescence at 504/ex and 523/em using a Tecan Infinite M1000 Pro plate reader. Data are normalized to vehicle control to avoid any differences in cell numbers between the groups.

**β secretase activity assay**. β-secretase activity was determined using fluorescent transfer peptides consisting of APP amino-acid sequences containing the cleavage sites of BACE secretase (BioVision). The method is based on the secretase-dependent cleavage of a secretase-specific peptide conjugated to the fluorescent reporter molecules EDANS and DABCYL, which results in the release of a fluorescent signal that was detected using a fluorescent microplate reader with excitation wavelength of 355 nm and emission at 510 nm. The level of secretase enzymatic activity is proportional to the fluorometric reaction, and the data is expressed as fold increase in fluorescence over that of background controls. BACE1 activity was assayed by a fluorescence-based in vitro assay kit according to the manufacturer's instructions (BioVision).

**Detection of protein aggregates**. For determination of misfolded protein aggregates, cells were fixed with 4% paraformaldehyde at RT for 15 min and, permeabilized in PBST (0.15% TritonX-100 in PBS) at RT for 15 min. Cells were then stained with proteostat aggresome detection dye at RT for 30 min and Hoechst 33342 nuclear stain, using the method described in the manual (Enzo Life Science Inc., Farmingdale, NY, USA). Proteostat (Enzo), a molecular rotor dye that becomes fluorescent when binding to the β-sheet structure of misfolded proteins. All components of the proteostat aggresome detection kit were prepared according to the manufacturer's instructions. Aggregated protein accumulation was detected using a Carl Zeiss 710 confocal microscope. (standard red laser set for the aggresome signal and DAPI laser set for the nuclear signal imaging). Further quantitative analyses, number of protein aggregates deposits per cell were counted.

**Quantification of mitochondrial copy number**. To quantify mitochondrial copy number, brain cortex tissue was homogenized, and total genomic DNA was isolated using the DNeasy blood and tissue kit (Qiagen 69504) according to manufacturer's instructions. Quantitative PCR (qPCR) quantification of mtDNA copy number was performed using the primers directed against the mitochondrial COXII gene and nuclear β-Globin using 100 ng of template DNA. For citrate synthase activity assays please see Supplementary Methods in Supplementary Information File.

**Statistics**. All results are presented as mean and ± SEM. Statistical analysis was performed using Prism 6.0 (Graph Pad Software). All experiments were replicated at least three times and measurements were taken on distinct samples. Individual data points mean and s.e.m. were displayed in the figures. Where appropriate column analyses were performed using an unpaired, two-tailed *t* test (for two groups) or one-way ANOVA with Bonferroni correction (for groups of three or more). For grouped analyses either multiple unpaired t-test with correction for multiple comparisons using the Holm–Sidak method or where appropriate two-way ANOVA with Tukey post hoc analysis was performed. $P$ values < 0.05 (95% confidence interval) were considered significant.

**Reporting summary**. Further information on research design is available in the Nature Research Reporting Summary linked to this article.

## Data availability
The data that support the findings of this study are available from the corresponding author upon reasonable request. The source data underlying Figs. 1–5 and Supplementary figures are provided as a Source Data file.

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

## Acknowledgements

We thank Trevor Tierney and Neil Shah for technical assistance with these studies. Support for this work was provided by grants from the NIH (R01HL136954, R01HL123966, 3R01HL123966-05S1, and R01HL142271 to J.W.E.,) and Pennsylvania Dept. of Health CURE Award (420792 to J.W.E.).

## Author contributions

J.W.E. conceived the project; J.W.E. and P.J. contributed to study design, data analysis, and writing the paper; P.J., D.T., and D.K., were involved with all assays and data collection; A.D.M. and D.K. were involved with all histology and memory test experiments; A.A.L. assisted with OxPhos studies; J.P.L and T.S.L. assisted with genetic mouse line characterization and breeding; D.P. provided human AD and 3xTg-AD mice samples and methodological expertize; J.W.E., D.P., and M.H.L. edited the manuscript and provided expertize with data interpretation.

## Additional information

**Competing interests:** The authors declare no competing interests.

