## [Peer Review File · Nature Communications]

Reviewers' comments:

Reviewer #1 (Remarks to the Author):

This study provides *in vivo* support for the mitochondrial calcium dyshomeostasis hypothesis for neurodegeneration. The studies are beautifully performed, using *in vivo* as well as *in vitro* methods to show that altering NCLX expression can modulate pathology and behavioral symptoms in the 3xTg-AD mouse model. The study involves an extensive amount of work, combining human tissue, cell culture and mouse data. With few exceptions noted below, the studies are solidly performed and technically sound. The data are significant in that they provide an *in vivo* extension of an existing body of work on mitochondrial calcium dysregulation in neurodegenerative diseases. Enthusiasm would be further increased if the current data were discussed mechanistically in the context of existing literature.

1. While technically elegant, the study marches through a long list of phenotypes known to be observed in AD models without providing much mechanistic integration.
 - a. How is NCLX expression and presumably, mitochondrial calcium, linked to the disparate phenotypes that are affected? What is the proposed sequence or proposed linchpin to the cascade?
 - b. The potential role of MCU dysregulation concurrent to the NCLX changes is not addressed experimentally or in discussion, despite significant changes in Fig. 1 and S1. Also, the characterization of MICU1 as a negative regulator of mtCU may be inaccurate/simplistic (Patron 2014).
 - c. What might underlie loss of NCLX as result of 3xTg mutations, and would similar mechanism apply to sporadic AD tissues?
2. The introduction and discussion seem too narrowly focused on AD. The authors have missed the chance to link their studies to broader themes in the larger literature. Also, there are too many reviews/opinion pieces cited up front resulting in omission of some foundational work in mitochondrial calcium regulation.
 - a. Ref. 21 may include the 2013 paper from the Rizzuto group that showed MICU1 acted as a gatekeeper.
 - b. The authors' prior work shows similar effects of NCLX deletion and overexpression on cardiac injury and this might be discussed.
 - c. Ref. 18 omits the Palty 2010 paper that established NCLX as the primary mechanism for mCa²⁺ efflux in excitable cells.
 - d. Other pertinent studies showing NCLX phosphorylation/activity in neuroprotection should be discussed (Kostic 2015, Verma 2017).
3. The culture experiments of Fig. 4 should show the Control + Ad-NCLX condition. One might expect NCLX overexpression could cause slightly diminished fxn as matrix calcium signals increase respiratory efficiency.
4. The data in Fig. 3N concerning AT180 and AT 270 should be re-examined. While the data concerning insoluble tau and AT8 is clear, the reported decrease in AT180 and AT270 is not apparent in the gel, and may result instead from increases to the 48 kDa HT7 band. Given that the phospho-tau is around 79 kDa band, the 79 kDa HT7 band, which does not appear to be changing, is the band that should be used for normalization.
5. In Fig 1A, two of the NCLX lanes and one control lane appear to show a technical problem (possible degradation postmortem?). Is it possible to show the Coomassie stain of the intact blot? Regardless, the densitometry and statistical analysis should be performed excluding these three strangely migrating lanes, and this should be mentioned.
6. Saturation of MCU and MICU2 blots might preclude meaningful densitometry, and the quality of some of the other blots could be improved. The densitometry for all figures should clearly state in either the y axis or the legend which specific band or bands were used for quantification and normalization when there is more than one band. An arrow pointing to bands of interest vs. nonspecific would also be helpful.

Minor.

7. Most assays are shown normalized to control with no information on the units for the original data and how data was transformed. Sometimes this makes it hard to understand exactly what is shown. Is Fig. 1H showing rates and what were the units? Peak amplitudes in 1G do not match 1F, so how were peaks defined/calculated? In Fig. 4P-R, instead of percent control, it would be better to display as percent of cells with cytox green.
8. For Fig. 3D-H – the text claims complete rescue. A discussion of why there was no rescue at 6 months (5 months after transgene activation) may be interesting.
9. What is rationale for switching back and forth between Tukey's and Sidak's post-hoc tests?

Reviewer #2 (Remarks to the Author):

Jadiya et al report the role of mitochondrial calcium efflux in Alzheimer's disease progression including amyloid pathology through neuronal specific mitochondrial Na⁺/Ca²⁺ exchanger (NCLX). Genetic deletion of neuronal specific mitoNCLX accelerated memory decline and increased amyloid and tau pathology in 3XTg AD mice. Restoration of NCLX rescued these defects. The outcomes of studies suggest the significance of mitoNCLX relevant to the AD pathology and mitochondrial function. However, the mechanistic aspects are weak. There are following concerns and need to have additional experiments to address these.

1. Some of immunoblots are poorly presented such as Fig. 1A, C that missed bands.
2. Although they analyzed the mitochondrial function in APP^{swe} cells, it is required for evaluation of mitochondrial function, permeability transition pore in vivo transgenic mice of NCLX/3xTg mice with deletion and restoration to provide direct evidence of effect of NCLX on AD-related mitochondrial properties. They also need to analyze mitochondrial properties including membrane potential and calcium retention ability from in vivo animal study.
3. For the detection of protein aggregation, it needs to clarify and characterize which protein aggregates and location. Are they related to Abeta or tau or others?
4. Characterization of NCLX mice. There is a lack of verification of neuronal localization of NCLX in mice
5. There are many grammatical errors

Reviewer #3 (Remarks to the Author):

Review

In this manuscript Jadiya et al report the role of mitochondrial loss of calcium flux in early stages of AD, provoked by progressive decreases in the expression of NCLX. Using cellular and animal models of AD, the authors show that genetic rescue of NCLX expression rescues significant AD molecular and behavioral phenotypes. The results presented in this report are certainly intriguing and demonstrate the pathological role of calcium deregulation in the disease. However, the author's conclusions cannot be fully supported without addressing some major points.

- 1) First and foremost, the cellular and animal AD models showed a clear effect on MICU1, MICUb and MICU2 expression that casts a shadow on the exclusive role of NCLX in the alteration of mitochondrial calcium homeostasis. Decreases in MICU1 have already been shown to impair mitochondrial calcium uptake (not rescued by overexpression of MICU2). However, despite these effects, mitochondrial calcium uptake seems not to be affected in their system. This result is highly controversial and should be addressed by the authors.
- 2) Nevertheless, it is quite challenging to pierce out the specific effects of NCLX in calcium regulation, as long as the expression of MICU1 is not restored in their system.
- 3) By the same token, the authors show that in their 3xTg-AD-NCLX model, NCLX is specifically reduced at 2 months old while the expression of MICUs is not affected. However, most behavioral results are performed at 6 months old. In order to understand the role of NCLX these behavioral

experiments should be performed at 2 months old. Likewise, WB showing the expression of MICUs at 6m, 9m and 12m are necessary to support their conclusions.

4) To understand the effects of rescuing NCLX expression on APP processing the authors should show the levels of C99 and C83 production. Similarly, the levels of amyloid beta should be reported as 42:40 ratio, which is the standard in the field.

5) Importantly, some discrepancies between the authors' conclusions and the quantifications of calcium dynamics should also be addressed. Specifically, in figure 1F the authors show their results on mCa⁺ amplitude in control and Swedish mutant cell lines before and after NCLX expression has been rescued. The quantification in figures G and H shows that NCLX expression only partially rescues this phenotype, which suggest that mitochondrial calcium uptake might be also impaired.

6) Seahorse analysis in figure 4 is not acceptable. First, the respiratory rate seems quite low for the number of cells assayed. In addition, the proton leak rate compared to basal respiration is too high. Moreover, there are clear discrepancies between the seahorse graphs and the quantifications. For example, while in figure 4b there is a clear difference in basal respiration between control and the mutant rescued by expression of NCLX figure 1C shows no difference. Equal discrepancies are shown in the quantification of the spare respiratory capacity. Similarly, the authors show in Fig 4D, that while ATP-linked respiration is significantly affected, mitochondria from cells carrying the Swedish mutation do not present with any alteration in coupling. Nevertheless, ATP-linked respiration should be quantified by calculation of the respiratory control index.

7) Finally, the authors should quantify mitochondrial mass by analyzing the ratio of mtDNA versus nDNA and/or PGC1alpha mRNA expression. Measuring the levels of mitochondrial proteins is not a reliable measure. In fact, while VDAC levels do not change, MICU1 expression is extremely affected. Moreover, the full-length blots from figure 2c suggest an altered expression of some OxPhos components.

Comment:

In some of the experiments presented in this manuscript, the concentrations of the reagents used are extremely high compared to the standards in the field. Specifically, FCCP, digitonin and succinate are used at a concentration 10 times higher compared to what is reported in the literature.

Response to Reviewers' Critique – Jadiya et al. Nat Comms. 2019

We would like to thank the editor and reviewers for their invaluable comments and suggestions. We have carried out numerous additional experiments to address the concerns raised and we believe we have greatly enhanced the quality and impact of the manuscript. We thank all three reviewers for the positive assessment of our work and constructive feedback.

Reviewer #1:

This study provides *in vivo* support for the mitochondrial calcium dyshomeostasis hypothesis for neurodegeneration. The studies are beautifully performed, using *in vivo* as well as *in vitro* methods to show that altering NCLX expression can modulate pathology and behavioral symptoms in the 3xTg-AD mouse model. The study involves an extensive amount of work, combining human tissue, cell culture and mouse data. With few exceptions noted below, the studies are solidly performed and technically sound. The data are significant in that they provide an *in vivo* extension of an existing body of work on mitochondrial calcium dysregulation in neurodegenerative diseases. Enthusiasm would be further increased if the current data were discussed mechanistically in the context of existing literature.

We greatly appreciate the positive feedback. Your suggested changes have greatly improved the quality of the manuscript and impact of our story. We have greatly added to the discussion to place our results within the context of existing literature.

Major concerns

1. While technically elegant, the study marches through a long list of phenotypes known to be observed in AD models without providing much mechanistic integration.

To our knowledge this is the first study to causatively examine mitochondrial calcium exchange in AD. Our data place impairments in mCa^{2+} efflux as a primary contributor of mitochondrial dysfunction and neuronal pathology in AD. We think our gain and loss-of-function models provide significant mechanistic data to suggest that mitochondrial calcium flux is indeed involved in the pathogenesis associated with these disease models. We have also added new studies to provide more 'mitochondrial' readouts of our *in vivo* phenotype with both loss of NCLX and gain-of efflux capacity including: mitochondrial content, calcium retention capacity experiments (MPTP activation), and *in vivo* mitochondrial superoxide production (please see **New Figure 5** and detailed response for reviewer # 2, Q#2 below). These data provide significantly more mechanistic integration of our findings regarding mitochondrial calcium transport.

2. How is NCLX expression and presumably, mitochondrial calcium, linked to the disparate phenotypes that are affected? What is the proposed sequence or proposed linchpin to the cascade?

It is widely accepted that dysregulation of intracellular calcium (iCa^{2+}) handling/elevation of iCa^{2+} contributes to AD pathology¹⁻⁵. Since mCa^{2+} exchange is an important regulator of cellular respiration and cell death, both of which are involved in AD pathogenesis it's intriguing to speculate that the central 'dysregulation' is at the level of mitochondrial calcium uptake and efflux. Indeed, our data suggest that both the mitochondrial calcium uniporter and NCLX expression are altered during AD progression. Our current working hypothesis is that mCa^{2+} overload, due to alterations in efflux, is an early pathogenic event, which contributes to increased ROS production and impaired mitochondrial energetics and subsequent neurodegeneration. It appears this pathogenic cascade precedes A β accumulation, tau phosphorylation and ultimately neurodegeneration/AD pathology. Our current working hypothesis is that some of the changes in mitochondrial calcium flux may be a compensatory attempt at elevating neuronal energetics very early on when there may be slight metabolic decompensation that is sensed by the mitochondria (perhaps Ca^{2+} -dependent dehydrogenases, etc)

and that this compensatory change quickly turns maladaptive in the high-intracellular calcium environment of AD pathogenesis. (Please see new discussion and citations in the revised manuscript.) The following schematic (**New Figure 6**) represents our best guess at the sequence of events and future studies will be aimed at testing other levels of this cyclic disease pathway.

3. The potential role of MCU dysregulation concurrent to the NCLX changes is not addressed experimentally or in discussion, despite significant changes in Fig. 1 and S1. Also, the characterization of MICU1 as a negative regulator of mtCU may be inaccurate/simplistic (Patron 2014).

We completely agree with the reviewer that MCU dysregulation in AD should be addressed further and we have added more to the results and discussion to highlight the changes at the level of the mitochondrial calcium uniporter (mtCU). Since the alteration of NCLX expression/activity is sufficient to revert the AD-associated phenotype we think the data presented here are sufficient to stand-alone. However, we also agree that there is sufficient remodeling of the mtCU, in addition to changes in NCLX, to warrant further study. Along these lines we have begun a series of extensive genetic experiments using our previously published MCU mutant mouse models crossbred with models of AD and hope to have these data for a followup manuscript. The extensive nature of modulating uptake in vivo in these aging studies will require a separate manuscript, as tackling the uptake pathway is beyond the scope of our focus here.

Also, Yes, we agree our MICU1 description was a bit simplistic and have changed this in the revised manuscript. "*MICU1 (an inhibitor of mtCU uptake at low iCa^{2+} and augments of mtCU uptake at high iCa^{2+} levels)*"⁶⁻⁹.

4. What might underlie loss of NCLX as result of 3xTg mutations, and would similar mechanism apply to sporadic AD tissues?

Our working hypothesis is that mitochondrial calcium efflux is impaired early on in AD and may be a driver of disease progression. We do not yet know the underlying mechanisms for loss of NCLX, but we think it's both at the transcriptional and protein degradation levels. Here we provide powerful evidence that the loss of NCLX precedes quantifiable neuronal pathology and cognitive decline in

these models and therefore this is not merely secondary to rampant neurodegeneration. As stated above, our current hypothesis is that there is a low-level metabolic stress^{10,11} that elicits remodeling of the mCa^{2+} exchange machinery to augment calcium uptake (increased matrix calcium content for activation of dehydrogenases) to increase mitochondrial energetics, which quickly turns maladaptive and leads to neuronal demise. We believe that similar mechanisms also apply to sporadic AD etiology, because we found loss of NCLX expression in non-familial AD tissue samples (Figure 1A).

5. The introduction and discussion seem too narrowly focused on AD. The authors have missed the chance to link their studies to broader themes in the larger literature. Also, there are too many reviews/opinion pieces cited up front resulting in omission of some foundational work in mitochondrial calcium regulation.

We agree with the reviewer and have now tried to cite original work and expanded the intro and discussion to include broader themes and highlight unifying theories of pathogenesis across model systems.

6. Ref. 21 may include the 2013 paper from the Rizzuto group that showed MICU1 acted as a gatekeeper.

We now include this citation at this location, in addition to the other places it's mentioned.

7. The authors' prior work shows similar effects of NCLX deletion and overexpression on cardiac injury and this might be discussed.

We agree and have now tried to draw parallels and contrasts with our work previously published work in Nature¹².

8. Ref. 18 omits the Palty 2010 paper that established NCLX as the primary mechanism for mCa^{2+} efflux in excitable cells.

We thank the reviewer for catching this oversight and have now cited this paper where appropriate.

9. Other pertinent studies showing NCLX phosphorylation/activity in neuroprotection should be discussed (Kostic 2015, Verma 2017).

We agree and these citations have been added to the discussion as requested.

10. The culture experiments of Fig. 4 should show the Control + Ad-NCLX condition. One might expect NCLX overexpression could cause slightly diminished fxn as matrix calcium signals increase respiratory efficiency.

We have now added the Ad-NCLX control in WT cells to all of our Seahorse experiments. We originally left it out because we did not see much of an impact on calcium exchange (**see Figure 1 F-N**). The Ad-NCLX control had little to no effect on OCR as shown in **New Figure 4 (B-G) and below**. We didn't find any OxPhos changes in the Con + Ad-NCLX group. This is not surprising since NCLX activity is dependent upon relative changes in Na^+ and Ca^{2+} flux and further our data in Figure 1 showed no effect on basal mito calcium content (Figure 1N).

11. The data in Fig. 3N concerning AT180 and AT 270 should be re-examined. While the data concerning insoluble tau and AT8 is clear, the reported decrease in AT180 and AT270 is not apparent in the gel and may result instead from increases to the 48 kDa HT7 band. Given that the phospho-tau is around 79 kDa band, the 79 kDa HT7 band, which does not appear to be changing, is the band that should be used for normalization.

We, as well as others, clearly detect an HT7 immunoreactive bands at ~75 kD (corresponding to full-length tau), as well as lower molecular weight band at ~50 kD. It has previously been suggested that unphosphorylated tau migrates faster at ~50 kD, while hyperphosphorylation slows tau migration to ~75kD¹³. One of our collaborators, an expert in an AD research, suggested to us to include both the bands for analysis of total tau expression¹⁴. However, as the reviewer requested, we have now quantified phospho AT180, AT270, AT8 and PHF-13 by correcting to the 79 kD HT7 band only. (Please see the **Supplementary Figure S7 Q'-T', and below**). We found a significant decrease in phospho-tau at S202/T205 (AT8 immunoreactivity), with no changes in T181 (AT270 immunoreactivity) and T231/ S235 (AT180 immunoreactivity) residues when quantified against the 75 kD HT7 band.

12. In Fig 1A, two of the NCLX lanes and one control lane appear to show a technical problem (possible degradation postmortem?). Is it possible to show the Coomassie stain of the intact blot? Regardless, the densitometry and statistical analysis should be performed excluding these three strangely migrating lanes, and this should be mentioned.

We apologize for the less than optimal blots and have carefully re-run Westerns for the frontal cortex samples acquired from non-familial AD patients and age-matched controls using another NCLX antibody that we recently confirmed with our genetic models (SLC24A6, abcam; ab83551). Please see **New Figure 1A and Supplementary Figure S6**, and below which clearly shows a decrease in NCLX expression. Densitometry and statistical analysis are shown in **Figure S7 and below**.

13. Saturation of MCU and MICU2 blots might preclude meaningful densitometry, and the quality of some of the other blots could be improved. The densitometry for all figures should clearly state in either the y-axis or the legend which specific band or bands were used for quantification and normalization when there is more than one band. An arrow pointing to bands of interest vs. nonspecific would also be helpful.

We have now re-bloted for MCU, MICU2 and MICU1 as requested, please see **New Figure 1C and below**. We also included an arrow pointing to the band of interest in **Figure S6** as requested. The indicated respective immunoreactive band was used for semi-quantification by densitometry. Please see full Western blots and analysis in **Supplementary Figure S6 & S7**.

Minor Concerns:

14. Most assays are shown normalized to control with no information on the units for the original data and how data was transformed. Sometimes this makes it hard to understand exactly what is shown. Is Fig. 1H showing rates and what were the units? Peak amplitudes in 1G do not match 1F, so how were peaks defined/calculated? In Fig. 4P-R, instead of percent control, it would be better to display as percent of cells with cytox green.

Fig. 1H, we now present the data as “ mCa^{2+} efflux rate” and units expressed as $\Delta F/sec$. mCa^{2+} efflux rate was calculated as $(F_{max}/F_0) - (F_{end}/F_0)/sec$. ΔF = change in fluorescence intensity, F_0 = fluorescence intensity at 0 sec, F_{max} = maximum fluorescence intensity, F_{end} = fluorescence intensity at the endpoint. Time was presented in sec. We have included this information in figure legends and methods.

Fig. 1G, peak amplitude was calculated as $(F_{max}/F_0) - (F_0/F_0)$. F_{max} = maximum fluorescence intensity, F_0 = fluorescence intensity at 0 sec.

In Fig. 4P-R, we evaluated membrane rupture using SYTOX Green. We seeded equal number of cells/groups and measured the fluorescence (504/ex, 523/em) in a standard plate reader assay. Data was normalized to vehicle control to avoid any differences in cell # between the groups. There is no readout for non-ruptured cells in this assay but we also provide data for cell viability in supplementary Figure S4.

All of these data have been updated in their respective figures and in the methods section.

15. For Fig. 3D-H – the text claims complete rescue. A discussion of why there was no rescue at 6 months (5 months after transgene activation) may be interesting.

It is interesting to note that neuronal NCLX expression (Camk2a-tTA x TRE-NCLX) rescued the age-associated deficits in contextual and cued recall in the 3xTg-AD model at 9 and 12 months, when AD pathology is progressive. In the 3xTg-AD mice model, significant impairments in contextual and cued recall were not observed until 9 mo. of age and NCLX did completely rescue this deficiency. However, as pointed out by the reviewer it is interesting that NCLX overexpression did not rescue the slight defect in spatial memory at 6 months of age but did at 9 and 12 months. We are hypothesizing that this is due to differences in neuronal sub-populations or brain regions (amygdala vs. hippocampus) and have added this to our discussion.

16. What is rationale for switching back and forth between Tukey's and Sidak's post-hoc tests?

Both methods are commonly used to compare among means of multiple groups. In our studies, we have used Sidak's post-hoc test, which uses pairwise multiple comparison test based on a t statistic and adjusts the significance level for multiple comparisons and provides tighter bounds (particularly appropriate for comparing vs. control). Tukey's multiple comparisons test were used only in 2 results including in Fig 1D and Fig 1N. Tukey's uses the studentized range distribution statistic to make all of the pairwise comparisons between groups. After consulting a statistician, we now report all results using the Sidak's post-hoc test analysis for consistency in data reporting.

Reviewer #2:

1. Some of immunoblots are poorly presented such as Fig. 1A, C that missed bands.

We apologize for the poor quality of some of the Westerns. NCLX is notoriously difficult to western blot for, likely because of the 13 transmembrane domains embedded in the IMM. We have rerun many of these as also requested by Reviewer #1 above in concerns # 12 and 13. Please see **New Figure. 1A, 1C** and below.

Figure. 1A

Figure. 1C

2. Although they analyzed the mitochondrial function in APPswe cells, it is required for evaluation of mitochondrial function, permeability transition pore in vivo transgenic mice of NCLX/3xTg mice with deletion and restoration to provide direct evidence of effect of NCLX on AD-related mitochondrial properties. They also need to analyze mitochondrial properties including membrane potential and calcium retention ability from in vivo animal study.

We agree with the reviewer and now provide data from these difficult experiments. As the reviewer has suggested we performed new experiments to evaluate mitochondrial function and MPTP opening using mitochondria isolated from our *in vivo* transgenic mouse models with NCLX deletion and restoration in the 3xTg background. Please see **New Figure 5 and below**. We performed calcium retention capacity assays (CRC, gold standard for MPTP susceptibility) using mitochondria isolated from the frontal cortex of 12 mo. old mice. Genetic deletion of neuronal NCLX in 3xTg-AD mice significantly decreased the CRC and increased susceptibility for permeability transition as compared to 3xTg-AD mice (**New Figure 5A-B and below**). Additionally, restoration of NCLX expression corrected the decrease in CRC seen in 3xTg-AD mice (**New Figure 5C-D and below**) and decreased susceptibility for MPTP opening. We think these valuable *in vivo* results greatly strengthen our overall hypothesis.

We next performed dihydroethidium (DHE) staining for *in vivo* detection of ROS production (primarily a superoxide detector). Cortical and hippocampal slices were freshly prepared from 12 mo. old mice

and stained with DHE. 3xTg-AD x NCLXcKO mice showed increased superoxide levels in both the cortex and hippocampus (Please see **New Figure 5E-G**). Rescue mCa^{2+} efflux in 3xTg-AD mice (NCLX expression) significantly decreased superoxide production (**New Figure 5H-J and below**).

In addition, we examined mitochondrial content using three different assays. We quantified the ratio of mtDNA versus nDNA, PGC1alpha mRNA expression and the citrate synthase activity to evaluate mitochondrial content in NCLX knockout and overexpressed 3xTg-AD mice at 2 & 12 mo. age. We didn't find any change in mitochondria mass within the groups (**New Figure 5K & L, Supplementary Figure S5 and below**), but it decreases with the ageing.

Combined, these results strongly suggest that NCLX dependent-efflux is a powerful modulator of mitochondrial function during AD disease progression.

3. For the detection of protein aggregation, it needs to clarify and characterize which protein aggregates and location. Are they related to Abeta or tau or others?

For *in vivo* detection of amyloid deposits and tau throughout the cerebral cortex and hippocampus, we performed Immunohistochemistry using A β -4G8 and the HT7 antibody, respectively. 4G8 antibody is reactive to amino acid residues 17-24 of β amyloid and the epitope lies within amino acids 18-22 of β amyloid. 4G8 β -amyloid antibody reacts to abnormally processed isoforms, as well as precursors. The epitope for HT7 lies within amino acids 159-163 of tau. For *in vitro* detection of protein aggregates, we employed a fluorescent dye (ProteoStat), which greatly increases fluorescence intensity upon binding to the β -sheet structure of misfolded proteins. All of these reagents are used extensively in the literature and have previously been reported. Detailed protocols and reagent information is provided in the Methods.

4. Characterization of NCLX mice. There is a lack of verification of neuronal localization of NCLX in mice.

In our study, we have crossed our NCLX conditional mutant mice (NCLX^{fl/fl})¹⁵ with a Camk2a-Cre (**neuronal-restricted Cre recombinase** transgenic model) to delete NCLX from the forebrain, specifically the prefrontal cortex and CA1 pyramidal cell layer in the hippocampus. These driver models have been extensively characterized and expression is limited to neurons¹⁶. Further, we report a significant reduction in hippocampal NCLX protein expression by Western blot (**Figure. 2C; and below**). Similarly, for the generation of neuronal specific NCLX overexpression, we crossed previously characterized mutant mice (NCLX-TRE¹²) with the Camk2a-tTA driver line (neuronal-restricted expression, doxycycline-off; Jackson Laboratory, stock no. 003010) transgenic model¹⁷. This model resulted in increased neuronal hippocampal NCLX protein expression as evaluated by Western blot analysis (**Figure. 3C; and below**). We also confirmed NCLX expression in our neuroblastoma N2a and APP^{swe} cells lines. The decrease in NCLX expression in APP^{swe} cells was completely rescued 48h post-infection with Adenovirus encoding NCLX (**Figure. 1E; and below**). In addition, multiple recent studies from multiple groups have confirmed mitochondrial localization of NCLX in neurons in addition to our extensive phenotyping in myocytes (see Luongo, Nature 2017, reports from Israel Sekler's group^{18,19} and also Andrey Abramov's group^{20,21}).

In addition, a recent transcriptome database by RNA sequencing shows distinct expression of NCLX (*Slc24a6*) in neurons²².

5. There are many grammatical errors.

We apologize and have carefully edited the manuscript.

Reviewer #3 (Remarks to the Author):

In this manuscript Jadiya et al report the role of mitochondrial loss of calcium flux in early stages of AD, provoked by progressive decreases in the expression of NCLX. Using cellular and animal models of AD, the authors show that genetic rescue of NCLX expression rescues significant AD molecular and behavioral phenotypes. The results presented in this report are certainly intriguing and

demonstrate the pathological role of calcium deregulation in the disease. However, the author's conclusions cannot be fully supported without addressing some major points.

We thank the reviewer for their positive appraisal of the manuscript and constructive critique.

1) First and foremost, the cellular and animal AD models showed a clear effect on MICU1, MICUb and MICU2 expression that casts a shadow on the exclusive role of NCLX in the alteration of mitochondrial calcium homeostasis. Decreases in MICU1 have already been shown to impair mitochondrial calcium uptake (not rescued by overexpression of MICU2). However, despite these effects, mitochondrial calcium uptake seems not to be affected in their system. This result is highly controversial and should be addressed by the authors.

We agree with the reviewer that MICU1, MICUb and MICU2 likely play important roles in mCa^{2+} homeostasis in AD. The loss of the key mCa^{2+} efflux mediator, NCLX, and reductions in negative regulators of mitochondrial calcium uniporter channel (mtCU): MICU1 and MICUb likely also contribute to promote mitochondrial calcium overload. These findings further support our hypothesis that mCa^{2+} overload is a key contributor to AD pathology and may contribute to metabolic deficits and neuronal demise. Here, mCa^{2+} overload appears to primarily be caused by impaired mCa^{2+} efflux because genetic rescue of neuronal NCLX in 3xTg-AD mice reduces pathogenic mCa^{2+} , AD-associated pathology and memory loss. We agree that there is sufficient remodeling of the mtCU, in addition to changes in NCLX, to warrant further study. Along these lines we have begun a series of extensive genetic experiments using our previously published MCU mutant mouse models crossbred with models of AD and hope to have these data for a followup manuscript. The extensive nature of modulating uptake *in vivo* in these aging studies will require a separate manuscript, as tackling the uptake pathway is beyond the scope of our focus here.

2) Nevertheless, it is quite challenging to pierce out the specific effects of NCLX in calcium regulation, as long as the expression of MICU1 is not restored in their system.

We respectively disagree, as enhancing efflux capacity should not in any direct way alter gating of the mtCU by MICU1. Again, in our model system the rescue of NCLX expression and function is enough to correct mCa^{2+} signaling defects and AD-associated pathology. Additionally, MICU1 expression was unchanged after deletion (**New Supplementary Figure S2 A-C**) or overexpression of NCLX (**Figure 3C**). This further suggests the observed phenotype is NCLX dependent and not MICU1. Restoration of NCLX expression corrected mCa^{2+} efflux, mitochondrial calcium retention capacity and reduced mCa^{2+} amplitude and matrix free- Ca^{2+} in APP^{swe} cell lines (Please see **Figure. 1F-N**). Similarly, restoration of NCLX expression in 3xTg-AD mice, increased the calcium retention capacity and decreased susceptibility for MPTP opening (**New Figure 5C & D**). In addition, enhancing mCa^{2+} efflux capacity rescued the AD-neuropathology and age-associated cognitive decline (Please see **Figure. 3**). These observations conclude that current study define the NCLX-dependent phenotype. These findings however do not rule out that modulating mitochondrial calcium uptake may likewise be a therapeutic approach to limit mitochondrial calcium overload in AD and we have future studies designed to address this.

3) By the same token, the authors show that in their 3xTg-AD-NCLX model, NCLX is specifically reduced at 2 months old while the expression of MICUs is not affected. However, most behavioral results are performed at 6 months old. In order to understand the role of NCLX these behavioral experiments should be performed at 2 months old. Likewise, WB showing the expression of MICUs at 6m, 9m and 12m are necessary to support their conclusions.

As the reviewer requested, we have performed behavioral experiments at 2 months old 3xTg-AD x NCLX-cKO mice (a time when no memory decline or pathology is noted in the 3xTg-AD model).

Please see **New Supplementary Figure S2 and below**. We found no cognitive impairments in the Y-maze spontaneous alteration and contextual and cued fear-conditioning test in 2mo old mice. These data are shown below and in the revised manuscript.

We also performed western blot showing the expression of MICUs and other mCa^{2+} exchanger at 2m, 9m and 12m. Please see **New Figure 2C and Supplementary Figure S2 and S7 and below**. It should be noted that we found a decrease in the expression of MICU1 in 3xTg-AD mice compared to age-matched non-transgenic controls (NTg), but MICUs expression was unchanged between 3xTg-AD and 3xTg-AD x NCLX-cKO mice. This new data is now included in the revised manuscript.

4) To understand the effects of rescuing NCLX expression on APP processing the authors should show the levels of C99 and C83 production. Similarly, the levels of amyloid beta should be reported as 42:40 ratio, which is the standard in the field.

We have tried numerous times with different antibodies to examine the levels of C99 and C83 production in the NCLX deleted/overexpressed 3xTg-AD mice but were unable to detect reliable expression of the α -secretase cleavage product, C83, or the β -secretase cleavage product, C99. We contacted two different labs with extensive experience with these pathways and both communicated that these are often difficult to detect.

However, in response to the reviewer's suggestion we now also report the levels of amyloid beta as 42:40 ratio. These data are presented in **Supplementary Figure S2, S3 and below**.

5) Importantly, some discrepancies between the authors' conclusions and the quantifications of calcium dynamics should also be addressed. Specifically, in figure 1F the authors show their results on mCa⁺ amplitude in control and Swedish mutant cell lines before and after NCLX expression has been rescued. The quantification in figures G and H shows that NCLX expression only partially rescues this phenotype, which suggest that mitochondrial calcium uptake might be also impaired.

We completely agree that NCLX expression partially rescues the impairments in mCa^{2+} handling in Fig 1F. However, we believe partial rescue of mCa^{2+} extrusion is sufficient to enhance the clearance of pathogenic mCa^{2+} . We found no major changes in mitochondrial calcium uptake (Please see **Supplementary Figure S1H**). We agree that targeting mitochondrial calcium uptake may be a viable therapeutic approach as well. The reviewer needs to understand that in this study we are only targeting efflux. Our lab and others have shown that increasing efflux does in no way limit mtCU-dependent uptake and we don't exactly understand the reviewer's criticism as enhancing efflux does not limit mtCU uptake.

6) Seahorse analysis in figure 4 is not acceptable. First, the respiratory rate seems quite low for the number of cells assayed. In addition, the proton leak rate compared to basal respiration is too high.

We performed numerous quality control experiments to make sure both cell number and drug concentrations were appropriate for these cells and our experimental system. It's unclear why the reviewer thinks this is quite low, given all the data is internally controlled and corrected to cell number (pmol/min/30k cells).

Moreover, there are clear discrepancies between the seahorse graphs and the quantifications. For example, while in figure 4b there is a clear difference in basal respiration between control and the mutant rescued by expression of NCLX figure 1C shows no difference. Equal discrepancies are shown in the quantification of the spare respiratory capacity. Similarly, the authors show in Fig 4D, that while ATP-linked respiration is significantly affected, mitochondria from cells carrying the Swedish mutation do not present with any alteration in coupling. Nevertheless, ATP-linked respiration should be quantified by calculation of the respiratory control index.

All of these data were calculated correctly, and we now supply all the raw values for the reviewer to reappraise these data independently. Quantification of basal respiration (base OCR – non-mito respiration (post-Rot/AA), ATP-linked respiration (post-oligo OCR – base OCR), Max respiratory capacity (post-FCCP OCR – post-Rot/AA), Spare respiratory capacity (post-FCCP OCR – basal OCR) and Proton leak (post-Oligo OCR – post Rot/AA OCR) was performed. To help appease concerns raised by reviewer #1's we have also checked OCR in the Control + Ad-NCLX condition. Please see the **New Figure 4B-G, and below for graph and data Table 1** for quantification of graph.

Data Table 1

Condition	Time (min)	Con		Con + Ad-NCLX		APPswe		APPswe + Ad-NCLX	
		Mean	SEM	Mean	SEM	Mean	SEM	Mean	SEM
Basal	X	77.89883	4.688615	75.91967	4.428652	60.27874	1.926857	71.34131	3.60726
Basal	7.966667	76.14039	4.535541	75.17449	4.399765	56.66433	2.053029	70.31419	3.287959
Post Oligo	14.63333	73.37215	4.426383	73.65749	4.296936	54.05029	2.2839	69.82796	3.073131
Post Oligo	21.53333	50.7214	2.172881	46.13517	2.243393	46.12835	0.790284	44.17177	1.630533
Post Oligo	28.15	50.18607	2.313717	46.2377	2.196464	44.35278	0.7272	44.0807	1.619611
Post FCCP	34.81667	49.34226	2.256019	45.4	2.139488	43.42432	0.655737	43.376	1.606152
Post FCCP	41.71667	105.0278	8.294305	106.7564	7.104289	74.56597	4.971664	99.0414	5.403469
Post FCCP	48.35	96.89738	7.476799	97.17896	6.223105	68.96401	4.389265	89.89222	4.583904
Post Rot/AA	54.96667	92.91283	7.29464	93.5216	6.019048	65.77766	4.257099	85.81026	4.337466
Post Rot/AA	58.66667	33.30798	1.334159	30.10582	2.672811	25.09793	0.602506	29.27971	1.433251
Post Rot/AA	65.28333	33.87932	1.379192	29.03521	1.252041	24.34719	0.771134	30.14697	1.227743
Post Rot/AA	71.95	34.4028	1.463805	29.60737	1.363328	24.5793	0.760467	29.90609	1.187163

7) Finally, the authors should quantify mitochondrial mass by analyzing the ratio of mtDNA versus nDNA and/or PGC1alpha mRNA expression. Measuring the levels of mitochondrial proteins is not a reliable measure. In fact, while VDAC levels do not change, MICU1 expression is extremely affected. Moreover, the full-length blots from figure 2c suggest an altered expression of some OxPhos components.

As requested we have examined mitochondrial mass by three independent methods including: mtDNA to nDNA ratio, PGC1alpha mRNA expression by qPCR and citrate synthase activity in both 2 & 12 month old mice from all control and experimental groups. Please see **NEW Figure 5K & 5L, Supplementary Figure S5 and below**. We found no difference in mtDNA/nDNA ratio, PGC1alpha mRNA expression and citrate synthase activity between the groups at either age. However, mitochondrial mass was reduced in the brains of 12 mo. old mice compared to 2 mo. old mice in all groups. These data suggest mitochondrial mass is unchanged with NCLX deletion or overexpression

in 3xTg-AD mice, but that mitochondrial mass is reduced with aging independent of mitochondrial calcium flux.

Quantification of mitochondrial mass

Minor Comments:

In some of the experiments presented in this manuscript, the concentrations of the reagents used are extremely high compared to the standards in the field. Specifically, FCCP, digitonin and succinate are used at a concentration 10 times higher compared to what is reported in the literature.

We have standardized the concentration of reagents based on cell type and assay conditions. We used 80- μ g/ml digitonin, 10- μ M succinate and 10- μ M FCCP. These concentrations have been reported in the literature^{23,24}. We have used much lower concentrations in other cell types and conditions.

Response to Critique-References

- 1 Begley, J. G., Duan, W., Chan, S., Duff, K. & Mattson, M. P. Altered calcium homeostasis and mitochondrial dysfunction in cortical synaptic compartments of presenilin-1 mutant mice. *Journal of neurochemistry* **72**, 1030-1039 (1999).
- 2 Mattson, M. P. *et al.* beta-Amyloid precursor protein metabolites and loss of neuronal Ca²⁺ homeostasis in Alzheimer's disease. *Trends Neurosci* **16**, 409-414 (1993).
- 3 Lopez, J. R. *et al.* Increased intraneuronal resting [Ca²⁺] in adult Alzheimer's disease mice. *Journal of neurochemistry* **105**, 262-271, doi:10.1111/j.1471-4159.2007.05135.x (2008).
- 4 Sepulveda-Falla, D. *et al.* Familial Alzheimer's disease-associated presenilin-1 alters cerebellar activity and calcium homeostasis. *The Journal of clinical investigation* **124**, 1552-1567, doi:10.1172/JCI66407 (2014).
- 5 Fedeli, C., Filadi, R., Rossi, A., Mammucari, C. & Pizzo, P. PSEN2 (presenilin 2) mutants linked to familial Alzheimer disease impair autophagy by altering Ca(2+) homeostasis. *Autophagy*, doi:10.1080/15548627.2019.1596489 (2019).
- 6 Patron, M. *et al.* MICU1 and MICU2 finely tune the mitochondrial Ca²⁺ uniporter by exerting opposite effects on MCU activity. *Molecular cell* **53**, 726-737, doi:10.1016/j.molcel.2014.01.013 (2014).
- 7 Csordas, G. *et al.* MICU1 controls both the threshold and cooperative activation of the mitochondrial Ca(2+)(+) uniporter. *Cell Metab* **17**, 976-987, doi:10.1016/j.cmet.2013.04.020 (2013).
- 8 Mallilankaraman, K. *et al.* MICU1 is an essential gatekeeper for MCU-mediated mitochondrial Ca(2+) uptake that regulates cell survival. *Cell* **151**, 630-644, doi:10.1016/j.cell.2012.10.011 (2012).

- 9 Liu, J. C. *et al.* MICU1 Serves as a Molecular Gatekeeper to Prevent In Vivo Mitochondrial Calcium Overload. *Cell reports* **16**, 1561-1573, doi:10.1016/j.celrep.2016.07.011 (2016).
- 10 Kapogiannis, D. & Mattson, M. P. Disrupted energy metabolism and neuronal circuit dysfunction in cognitive impairment and Alzheimer's disease. *Lancet Neurol* **10**, 187-198, doi:10.1016/S1474-4422(10)70277-5 (2011).
- 11 Yao, J. *et al.* Mitochondrial bioenergetic deficit precedes Alzheimer's pathology in female mouse model of Alzheimer's disease. *Proceedings of the National Academy of Sciences of the United States of America* **106**, 14670-14675, doi:10.1073/pnas.0903563106 (2009).
- 12 Luongo, T. S. *et al.* The mitochondrial Na⁽⁺⁾/Ca⁽²⁺⁾ exchanger is essential for Ca⁽²⁺⁾ homeostasis and viability. *Nature* **545**, 93-97, doi:10.1038/nature22082 (2017).
- 13 Sahara, N. *et al.* Assembly of tau in transgenic animals expressing P301L tau: alteration of phosphorylation and solubility. *Journal of neurochemistry* **83**, 1498-1508 (2002).
- 14 Joshi, Y. B. *et al.* Absence of ALOX5 gene prevents stress-induced memory deficits, synaptic dysfunction and tauopathy in a mouse model of Alzheimer's disease. *Human molecular genetics* **23**, 6894-6902, doi:10.1093/hmg/ddu412 (2014).
- 15 Luongo, T. S. *et al.* The mitochondrial Na⁺/Ca²⁺ exchanger is essential for Ca²⁺ homeostasis and viability. *Nature* **545**, 93-97, doi:10.1038/nature22082 (2017).
- 16 Tsien, J. Z. *et al.* Subregion- and cell type-restricted gene knockout in mouse brain. *Cell* **87**, 1317-1326 (1996).
- 17 Mayford, M., Bach, M. E. & Kandel, E. CaMKII function in the nervous system explored from a genetic perspective. *Cold Spring Harbor symposia on quantitative biology* **61**, 219-224 (1996).
- 18 Kostic, M. *et al.* PKA Phosphorylation of NCLX Reverses Mitochondrial Calcium Overload and Depolarization, Promoting Survival of PINK1-Deficient Dopaminergic Neurons. *Cell reports* **13**, 376-386, doi:10.1016/j.celrep.2015.08.079 (2015).
- 19 Sharma, V., Roy, S., Sekler, I. & O'Halloran, D. M. The NCLX-type Na⁽⁺⁾/Ca⁽²⁺⁾ Exchanger NCX-9 Is Required for Patterning of Neural Circuits in *Caenorhabditis elegans*. *The Journal of biological chemistry* **292**, 5364-5377, doi:10.1074/jbc.M116.758953 (2017).
- 20 Ludtmann, M. H. R. *et al.* LRRK2 deficiency induced mitochondrial Ca⁽²⁺⁾ efflux inhibition can be rescued by Na⁽⁺⁾/Ca⁽²⁺⁾/Li⁽⁺⁾ exchanger upregulation. *Cell Death Dis* **10**, 265, doi:10.1038/s41419-019-1469-5 (2019).
- 21 Gandhi, S. *et al.* PINK1-associated Parkinson's disease is caused by neuronal vulnerability to calcium-induced cell death. *Molecular cell* **33**, 627-638, doi:10.1016/j.molcel.2009.02.013 (2009).
- 22 Zhang, Y. *et al.* An RNA-sequencing transcriptome and splicing database of glia, neurons, and vascular cells of the cerebral cortex. *The Journal of neuroscience : the official journal of the Society for Neuroscience* **34**, 11929-11947, doi:10.1523/JNEUROSCI.1860-14.2014 (2014).
- 23 Antony, A. N. *et al.* MICU1 regulation of mitochondrial Ca⁽²⁺⁾ uptake dictates survival and tissue regeneration. *Nat Commun* **7**, 10955, doi:10.1038/ncomms10955 (2016).
- 24 Paillard, M. *et al.* Tissue-Specific Mitochondrial Decoding of Cytoplasmic Ca⁽²⁺⁾ Signals Is Controlled by the Stoichiometry of MICU1/2 and MCU. *Cell reports* **18**, 2291-2300, doi:10.1016/j.celrep.2017.02.032 (2017).

REVIEWERS' COMMENTS:

Reviewer #1 (Remarks to the Author):

The authors have addressed the majority of my concerns. While I would have liked to see additional mechanistic considerations for how changes in mitochondrial calcium result in Abeta and tau pathology, strengths of the study include use of multiple model systems to highlight a pathological role for mitochondrial calcium dyshomeostasis in AD. The observed downregulation of NCLX expression in AD, and in vivo impact are novel and add support to prior studies showing neuroprotective effects of NCLX phosphoactivation in PD models.

Reviewer #2 (Remarks to the Author):

Authors have adequately addressed this reviewer's concerns.

Reviewer #3 (Remarks to the Author):

The revised manuscript has improved substantially. I thank the authors for their effort on answering all the reviewers' requests. Although many of the concerns raised in the previous revision have been addressed, some of the author's conclusions cannot be supported by the data. I understand the limitations of the experimental approaches, and I agree that the authors are aiming at calcium efflux. However, the data presented here does not preclude that the behavioral phenotype detected is driven by alterations in MICU(s), rather than defects in NCLX, despite the beneficial effect of correcting NCLX levels.

Nevertheless, the data presented is certainly intriguing and I agree that further research out of the scope of this paper is necessary.

While no more experimental data is asked, I would suggest toning down some of the conclusions as well as the title of the manuscript to better match and reflect the data presented.

For instance, the title should not present impairments in calcium efflux as drivers of pathology since this is not what the authors have demonstrated.

In the discussion, the authors mention that : "no studies have examined the role of mitochondrial signaling in AD or have address it". This is certainly not the case as multiple reports have addressed this issue in the course of the disease. In fact, some precious articles have rescued mitochondrial dysfunction in AD by restoring other pathways.

In summary, I believe the data presented here are relevant, but given the limitations of this study, the conclusions should be more accurate, especially for a high impact journal such as Nature Communications.

Response to Reviewers' Critique – Jadiya et al. Nat Comms. 2019

We thank all three reviewers for the positive assessment of our work.

REVIEWERS' COMMENTS:

Reviewer #1 (Remarks to the Author):

The authors have addressed the majority of my concerns. While I would have liked to see additional mechanistic considerations for how changes in mitochondrial calcium result in Abeta and tau pathology, strengths of the study include use of multiple model systems to highlight a pathological role for mitochondrial calcium dyshomeostasis in AD. The observed downregulation of NCLX expression in AD, and in vivo impact are novel and add support to prior studies showing neuroprotective effects of NCLX phosphoactivation in PD models.

Response: We greatly appreciate your positive feedback.

Reviewer #2 (Remarks to the Author):

Authors have adequately addressed this reviewer's concerns.

Response: We thank the reviewer for their positive appraisal of the manuscript.

Reviewer #3 (Remarks to the Author):

The revised manuscript has improved substantially. I thank the authors for their effort on answering all the reviewers' requests. Although many of the concerns raised in the previous revision have been addressed, some of the author's conclusions cannot be supported by the data. I understand the limitations of the experimental approaches, and I agree that the authors are aiming at calcium efflux. However, the data presented here does not preclude that the behavioral phenotype detected is driven by alterations in MICU(s), rather than defects in NCLX, despite the beneficial effect of correcting NCLX levels. Nevertheless, the data presented is certainly intriguing and I agree that further research out of the scope of this paper is necessary.

We appreciate your statements and agree that mitochondrial calcium uniporter-dependent calcium uptake make also play a role in AD development and progression and have included this notion in the Results and Discussion as previously requested.

While no more experimental data is asked, I would suggest toning down some of the conclusions as well as the title of the manuscript to better match and reflect the data presented. For instance, the title should not present impairments in calcium efflux as drivers of pathology since this is not what the authors have demonstrated. In the discussion, the authors mention that :“no studies have examined the role of mitochondrial signaling in AD or have address it”. This is certainly not the case as multiple reports have addressed this issue in the course of the disease. In fact, some precious articles have rescued mitochondrial dysfunction in AD by restoring other pathways. In summary, I believe the data presented here are relevant, but given the limitations of this study, the conclusions should be more accurate, especially for a high impact journal such as Nature Communications.

Response: We thank the reviewer for their praise of our work and we understand the limitations of the experimental approaches. Your comments have helped us formulate clearer more concise discussion of our work. As requested we have toned down the rhetoric throughout the Results and Discussion and have modified the title, as you and the editor requested, to lessen our claims. Thank you for your critical review of our work.